**DOI: 10.1038/ncomms14306**　　**OPEN**

# MetaSort untangles metagenome assembly by reducing microbial community complexity

Peifeng Ji[1],[*], Yanming Zhang[1],[*], Jinfeng Wang[1] & Fangqing Zhao[1]

Most current approaches to analyse metagenomic data rely on reference genomes. Novel microbial communities extend far beyond the coverage of reference databases and *de novo* metagenome assembly from complex microbial communities remains a great challenge. Here we present a novel experimental and bioinformatic framework, metaSort, for effective construction of bacterial genomes from metagenomic samples. MetaSort provides a sorted mini-metagenome approach based on flow cytometry and single-cell sequencing methodologies, and employs new computational algorithms to efficiently recover high-quality genomes from the sorted mini-metagenome by the complementary of the original metagenome. Through extensive evaluations, we demonstrated that metaSort has an excellent and unbiased performance on genome recovery and assembly. Furthermore, we applied metaSort to an unexplored microflora colonized on the surface of marine kelp and successfully recovered 75 high-quality genomes at one time. This approach will greatly improve access to microbial genomes from complex or novel communities.

[1] Computational Genomics Lab, Beijing Institutes of Life Science, Chinese Academy of Sciences, Beijing 100101, China. * These authors contributed equally to this work. Correspondence and requests for materials should be addressed to F.Z. (email: zhfq@biols.ac.cn).

Currently, state-of-the-art metagenomic data analysis approaches largely rely on comparisons to reference genomes. However, these methods are of limited application because of the small fraction of reference genomes presented. The uncultured and unsequenced microbial majority, referred to as 'dark matter', constitutes at least 60 major lines of descent (phyla or divisions) within the bacterial and archaeal domains[1]. The situation is even more fundamentally skewed considering the bias that greater than 88% of all microbial isolates represent only four bacterial phyla[1]. Moreover, due to clonal differences, environmental adaptation or possible artefacts from cultivation processes, bacterial genomes from different isolates of the same species typically exhibit considerable genetic heterogeneity when compared[2]. Therefore, reference genomes and methods relying on these genomes place limitations on the discovery of previous unknown species. In particular, these practices limit our ability to understand the taxonomic composition and functional potential of novel microbial communities.

*De novo* metagenomic assembly has proven difficult due to the inherent complexities of microbial communities, including repeat sequences, uneven coverage and intra-species divergence[3–5]. A reasonable solution involves clustering these fragmented contigs into discrete units, referred to as 'binning'. When closely related reference genomes are lacking, binning must be performed in an unsupervised fashion. Numerous unsupervised binning methods that employ sequence compositions have been developed, but these methods only work well with extreme base compositions and fail to clearly separate taxonomically related organisms[6,7]. An alternative approach involves the use coverage patterns across multiple samples, allowing binning at the species level and occasionally the strain level[6,8,9]. These methods hold great promise for improving binning performance, but they require a large number of samples. Moreover, most binning methods only consider large contigs (typically ≥2 Kbp (refs 6,8,10)), which may not be applicable to most moderate- or low-abundance species in various microbial communities.

As a complement to classical metagenomics, single-cell sequencing, typically employing multiple displacement amplification (MDA) to amplify genomic DNA, has emerged as a powerful approach to target coherent biological entities[1,11]. In particular, compared with metagenomics, single-cell sequencing is more assessable to the genomic heterogeneity of target populations. However, single-cell sequencing demands a highly specialized laboratory facility, and whole-genome amplifications performed one at a time yield highly uneven sequencing depth and elevated levels of chimeric reads[12–14]. For example, Marcy *et al.* found that 82% of single-cell MDA reads were assembled into contigs, but less than 39% of the reference genome could be covered[15]. In addition, there is a high failure rate associated with this methodology. Rinke *et al.* sequenced 9,600 single cells and yielded 3,300 (34%) successful amplifications, among which only 201 (2%) draft genomes were recovered[1]. It is evident that sequencing single cells at a large scale is prohibitively time consuming, labour intensive and expensive.

To overcome this issue, McLean *et al.*[16] developed a new approach that involves forming random pools of single flow-sorted cells and sequencing all the cells (called a mini-metagenome) simultaneously. The mini-metagenome has higher throughput than the single-cell genome and lower complexity than the original metagenome. Moreover, an assembly tool, SPAdes[17], was designed for coping with wide variations in coverage and chimeric reads from MDA samples, and also it should be noted that these drawbacks of MDA could be resolved using newly developed low input DNA library preparation methods[18]. This strategy effectively increases the likelihood of capturing and assembling the genomes of low-abundance microorganisms. However, the mini-metagenome approach still faces two inherent challenges. (1) Without an efficient method to control the number and type of separated cells, different mini-metagenomes may share a large proportion of microorganisms, for example, 61.9% of species were shared by the two samples in their study. Such taxonomic overlap unavoidably results in some species being captured repetitively and thereby reduces the throughput and efficiency of the approach. (2) In each mini-metagenomic sample, only a few dominant species can be easily assembled. For low-abundance species, the problem of recovering complete genomes remains unresolved. This situation is even worse when considering the amplification bias of MDA.

To address these challenges and increase the utility of the mini-metagenome approach, we present metaSort, an efficient and high-throughput method that combines the advantages of traditional metagenomics and single-cell genomics to recover nearly complete genomes from metagenomic samples. Compared with previous approaches, metaSort can segregate small subsets of cells with minimum overlaps from the original metagenomic sample by sorting and gating the cells by size or other physical and chemical properties using flow cytometry (FCM). In addition, we propose two novel algorithms, Binning and Fishing (BAF) and Machine Learning and Graph-based Algorithm (MGA), which can assemble genomes based on the combination of the sorted mini-metagenome and the original metagenome. BAF is designed to generate merged genome bins from mini-metagenomic sequences and their connected contigs from the original metagenome. The resulting bins are employed by MGA to recover whole target genome sequences from the original metagenomic contig connection graph. Furthermore, MGA assembles the recovered target genome sequences into longer scaffolds and identifies variations. By testing on synthetic and real data sets, we demonstrated that metaSort can significantly improve metagenome assembly and recover strain-level variation profiles from complex microbial communities.

## Results

**The metaSort approach.** First, metagenomic DNA is extracted and sequenced, and the resulting reads are assembled into contigs using SOAPdenovo2 (ref. 19), which are called meta-O because they represent the original metagenomic sample. These contigs have a high genome coverage rate but are often fragmented due to the assembly complexity. Second, FCM is utilized to reduce the metagenome complexity by sorting the microbial cells and gating out small subsets including a specific number of target cells. For each subset, DNA is extracted, amplified by MDA, sequenced and assembled using SPAdes[12]. The assembled contigs of each subset, termed meta-S, are longer than those of meta-O. However, obtaining complete genome sequence for a specific taxon remains impossible due to the unevenness of MDA amplification. Hence, third, we employ a combination strategy that incorporates the advantages of both meta-O and meta-S to recover nearly complete genomes and detect variations using the BAF and MGA algorithms (Fig. 1). The algorithms are implemented in Python and available online as a free open-source tool (https://sourceforge.net/projects/metasort/).

The BAF algorithm is designed to cluster assembled contigs into bins, with each bin representing one target genome (Supplementary Fig. 1). The SPAdes-assembled contigs are first clustered into meta-S bins using the tetra-nucleotide frequency (TNF) features[8,10,20,21]. Then, to improve the binning efficiency, meta-S bins are mapped to the meta-O contig connection graph, which is built by creating links between

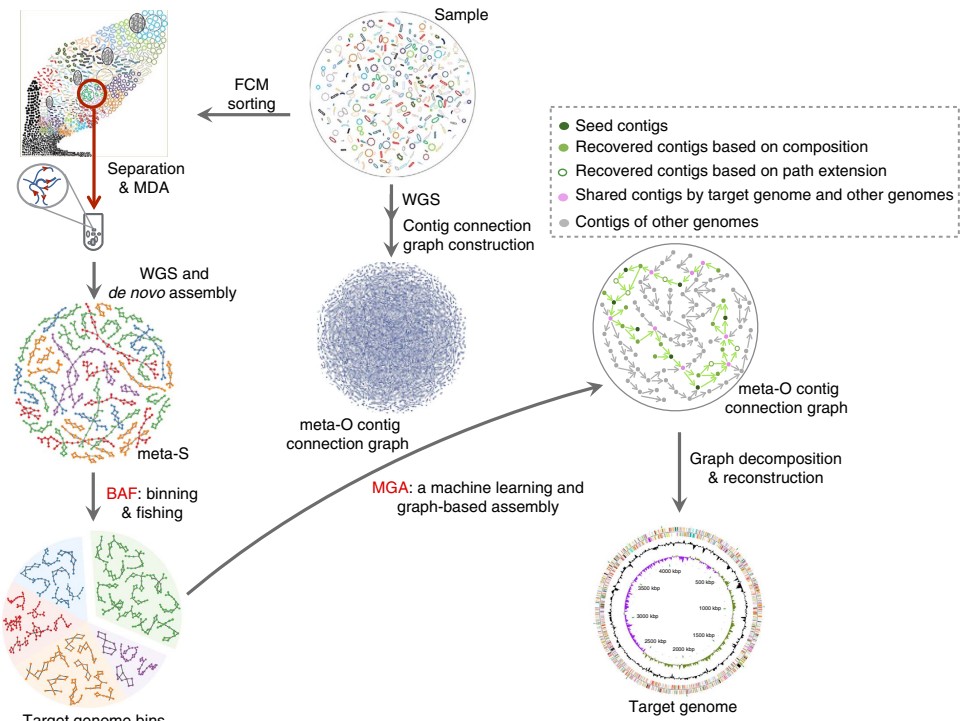

**Figure 1 | Overview of the metaSort approach.** First, metagenomic DNA is extracted and sequenced, and the resulting reads are assembled into contigs. Subsequently, these contigs, termed as meta-O, are used for constructing a contig connection graph, where nodes represent contigs and edges are created between two overlapped contigs. Second, the FCM is utilized to segregate small subsets of cells by gating microbial cells by size or other properties. For each subset, DNA is extracted, amplified by MDA and then sequenced. The resulting reads are assembled using SPAdes assembler. The assembled contigs of each subset, termed meta-S, are clustered into target genome bins using the BAF algorithm. Third, the MGA algorithm is performed to recover the remaining target contigs that missed by BAF based on meta-O contig connection graph and to assemble the target contigs into scaffolds.

two overlapping meta-O contigs. For those meta-S bins located on the meta-O contig connection graph, paths linking two bins are identified. Subsequently, the meta-S bins and the paths linking meta-S bins are extracted to form a new graph. Finally, this graph is partitioned into components based on graph connectivity and the meta-S bins in the same component are clustered into merged bins. The merged bins and the remainder of the meta-S bins that do not participate in the process of merging constitute the target genome bins.

Next, the MGA algorithm is performed to recover the remaining contigs of the target genome that are missed by BAF and to assemble the target contigs into larger scaffolds (Supplementary Fig. 2). This algorithm is a three-step process that includes increase, decrease and connection steps. Due to the low genome coverage rate of the target genome bins derived from the meta-S assembly, the increase step is performed to recover the remaining target contigs from meta-O. This step is achieved using a supervised support vector machine (SVM (ref. 22)), which requires positive, negative and test datasets for classification. MGA takes the sequences in the target genome bin as a positive data set, and the other two data sets are obtained using a distance-based method. After obtaining these data sets, SVM trains a classification model by taking the codon usage of contigs as feature vectors to predict target contigs. Due to the high false-positive rate of SVM, the decrease step is employed to remove contamination from the candidate target contigs using sequencing depth. In the connection step, both the remainder of candidate target contigs and seed contigs, referred as to landmarks, are used for traversing the meta-O contig connection graph. MGA starts at each landmark and searches against the meta-O contig connection graph to obtain the paths connecting

two landmarks. Once all the landmarks are visited on the meta-O contig connection graph, the landmarks along the searched paths are extracted to form a target genome contig graph. Algorithms are then applied to remove tips, merge bubbles and assemble the remaining contigs into scaffolds. Based on the bubble structure and component information in the connection step, MGA performs *de novo* variation detection. MGA first extracts the bubbles that are mainly caused by genetic variants by filtering the bubbles based on sequence divergence and sequencing depth. After calling variations using these filtered bubbles, we have proposed two metrics, bubble density and bubble identity, to assess the polymorphic sites in each component of the target genome contig graph. Then, both metrics are used as feature vectors of a logistic regression model to predict the strain-level variation in the target genome.

**MGA can recover nearly whole genome based on partial sequences.** To validate our computational algorithm, we constructed a simulated metagenomic data set consisting of 100 genomes with different sequencing depths ranging from 5× to 128× (Supplementary Fig. 3 and Supplementary Table 1). Ten species containing 2∼4 different strains or subspecies were designed to test the capability of MGA to recognize strain-level genomic variation. Reads were assembled, and the sources of contigs were then identified by mapping them back to the reference genomes. For each genome, 40% of the assembled contigs were randomly selected as 'seeds', and the sum length of these seed contigs represented 30-59% of the total genome coverage. We ran MGA to recover the remaining genome sequences based on these seeds and compared the results with their original reference genomes.

As shown in Supplementary Fig. 4, MGA successfully recovered most of the remaining sequences with an average genome coverage of 88.7% and a low contamination rate (an average of 0.6%). Among the assemblies, only two had a final genome coverage under 70%. Whereas both of these genomes contained an initial genome coverage $< 30\%$, we set the recommended initial genome coverage to 30% in the downstream analyses. We further compared recovered genome quality with three binning methods, CONCOCT (ref. 8), metaBAT (ref. 10) and MaxBin (ref. 23). As shown in Supplementary Fig. 5a, metaSort exhibited a substantial improved genome coverage over the other three binning methods ($P < 1 \times 1^{-10}$, $t$-test). We further compared their performance on the assembled contig length. As shown in Supplementary Fig. 5b, metaSort showed considerably improved N50 length compared with the other three binning methods. The average N50 length were 37-, 45- and 51-fold longer than that of MaxBin, metaBAT and CONCOCT, respectively. The performance of the BAF algorithm was also validated using the meta-O assembly. After binning, the average contamination rate was estimated to be 0.61% across 99% of the recovered genomes (Supplementary Fig. 6). Next, the efficiency of the increase and decrease steps in MGA were explored. As shown in Supplementary Fig. 4b, the increase step based on SVM exhibited high sensitivity (an average of 79.2%) but low precision (an average of 14.4%). After the decrease step, however, the precision was almost perfect at an average of 99.2% and was accompanied by a slight decrease in sensitivity (an average of 75.0%).

We further tested whether MGA could identify the species that contain strain-level variation. For each genome, the target genome contig graph was first partitioned into components based on graphic connectivity, and then both bubble density and bubble identity were calculated. As shown in Supplementary Figs 4c,7 of ten subspecies- or strain-containing genomes exhibited distinct patterns on both bubble density and bubble identity plots compared with other species. The most remarkable feature was that the species containing multiple strains had a significantly larger number of components than other species ($P = 2.2 \times 10^{-12}$, Wilcoxon rank-sum test), and both the bubble density and bubble identity metrics exhibited increased variation compared with the others. It should be noted that *Sulfolobus acidocaldarius*, *Listeria monocytogenes* and *Alteromonas macleodii* did not exhibit any obvious patterns on either bubble density or bubble identity because the strains in these three species exhibited a considerably increased level of average nucleotide identity (ANI) ($> 99.9\%$) compared with all other genomes (maximum of 99.52%). This finding indicated a threshold of similarity of 99.5% for ANI, above which genomes containing strain-level variations may not be identified. Similar results were also reported by Luo *et al.*[24] who found that related genomes with greater than 99.5% ANI could not be separated but were clustered into a single group.

**Segregating meta-O into low-complex subsets**. To examine whether metaSort provides an effective method to decrease the complexity of the original metagenome, we applied metaSort to well-characterized human salivary microbiota. Using different sized polystyrene beads as a size control (Supplementary Fig. 7a–d), four sets of meta-S, named S1 to S4, were isolated; the number of cells was $5.9 \times 10^4$, $2.8 \times 10^4$, $3.7 \times 10^4$ and $1.0 \times 10^5$, respectively (Supplementary Fig. 7d). Genomic DNA was extracted for each subset and then amplified by MDA. The amplified DNA was used to construct DNA libraries and then sequenced. The resulting reads of each meta-S were aligned with the NR database using BLASTX. Subsequently, the taxonomic profiles of each meta-S were annotated using MEGAN (ref. 25)

(Fig. 2a). As expected, the four sets of mini-metagenomes exhibited distinct taxonomic profiles, with *Pseudomonas* and *Enterobacter*, *Neisseria* and *Prevotella*, *Veillonella*, *Actinomyces* and *Rothia* as the most abundant genera in S1, S2, S3 and S4, respectively. Notably, low-abundance species in the original sample (meta-O) could be captured by our approach. For example, *Enterobacter cloacae* and *Enterobacter hormaechei* were enriched in S1 but were represented at low abundance in meta-O. Moreover, we observed that several species were shared by S1 and S2 but showed different relative abundances in the two mini-metagenomes, which was likely caused by the adjacent location of these two subsets during the FCM analysis. This finding highlights a more general sense that different regions were dominated by distinct bacteria after sorting by FCM.

**MGA recovers genomes by combining meta-S and meta-O**. The four mini-metagenome data sets of meta-S were assembled using the SPAdes assembler[12], yielding 50,325 scaffolds (N50 length of 6,877 bp). Moreover, the original salivary sample was sequenced using the Illumina HiSeq2000 PE 100-bp platform, and 142 million paired-end reads were generated with a median insert size of 180 bp. This meta-O data set was assembled using SOAPdenovo2 and generated 2,836,169 contigs, with N50 contig length of 250. The N50 length of the meta-S assembly was $27.5 \times$ larger than that of the meta-O assembly, indicating that the improved assembly largely benefited from the reduced complexity of meta-S. Then, the meta-S contigs were mapped back to the reference genomes to determine the coverage of genomes and the efficiency of MDA amplification. As expected, in contrast to the metagenomic sequencing of meta-O, the MDA-amplified meta-S samples exhibited highly uneven sequencing depth and thus led to low genome coverage (3.5-88.2%) (Fig. 2a). *Neisseria flavescens* and *Pseudomonas aeruginosa* serve as examples (Supplementary Fig. 8a,b). When mapping MDA-amplified reads to the reference genomes, extremely non-uniform sequencing depths through the chromosome were observed. Similar observations were noted in *Veillonella atypical*, *Streptococcus parasanguinis*, *Actinomyces graevenitzii* and *Neisseria mucosa* (Fig. 2a). Because most of the enriched species exhibited poor genome coverage, meta-S reads alone were not sufficient for reconstructing the complete genomes.

The assembled sequences of meta-S were clustered into target genome bins using BAF and the performance of this approach was evaluated by estimating the accuracy of resulting bins, which was referred from mapping these bins to their references. In all, 71.4% of the target genomes bins presented a contamination rate of zero and all other genomes have a contamination rate no more than 1.75%, except one was 53.37%, indicating the high accuracy of the BAF algorithm. For each bins that had at least 30% of total genome coverage in meta-S, we applied the MGA algorithm to recover the remaining genome sequences from meta-O and to make a combined assembly. Seven genomes were successfully recovered and assembled, where the genome coverage reached 82.0-97.6%, and the assembly quality was significantly improved with a 2.3- to 500.6-fold increase in NGA75 contig length (Supplementary Table 2), which is defined as the value N such that 75% of the finished sequence is contained in contigs whose alignments to the finished sequence are of size N or larger. Furthermore, the bins had genome coverage below 30% in meta-S also evaluated to validate the performance of MGA on low coverage genomes. As shown in Supplementary Table 3, both the contig length and genome coverage exhibited considerable enhancement (Supplementary Table 3). Specifically, the average genome coverage increased from 24.6 to 79.0%.

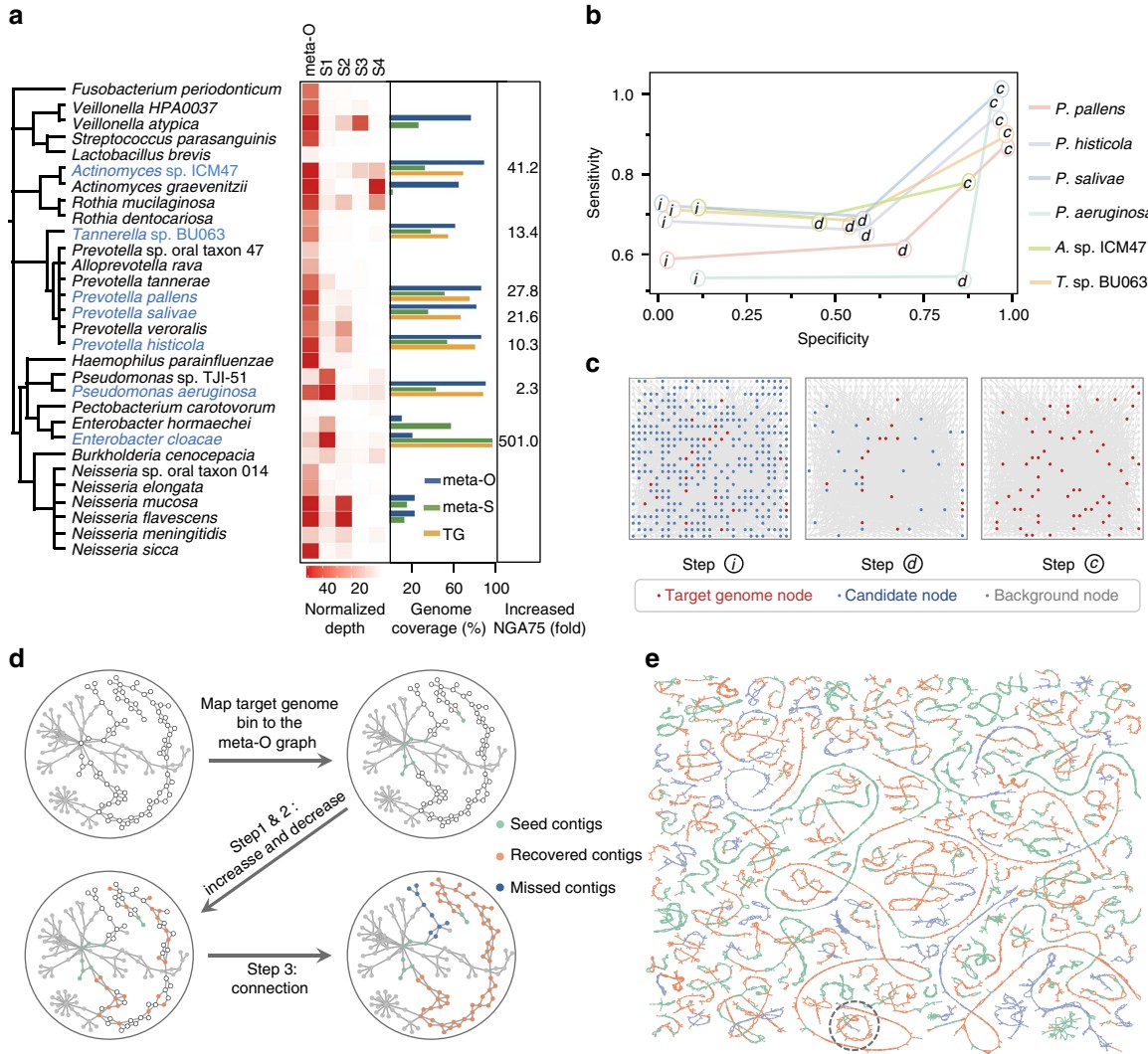

**Figure 2 | Application of metaSort on an oral microbiome. (a)** MGA assembly of the genomes enriched in meta-S. Heatmap shows the normalized sequencing depth of the genomes in four mini-metagnomes data sets of meta-S (S1-S4) and meta-O. Bar-plot represents the genome coverage in the meta-O, meta-S and MGA assemblies. The increased NGA75 after MGA assembly is shown in the right panel. **(b)** Sensitivity and specificity of target contig recovery in the three key steps of MGA. '*I*', '*d*' and '*c*' represents the increase, decrease and connections step, respectively. **(c)** Target contig recovery and contaminated contig filtering during the increase, decrease and connection steps, as illustrated by using a component of *Pseudomonas aeruginosa* contig connection graph. **(d)** A component of *Prevotella salivae* contig connection graph is extracted to illustrate the increasing process of target contig recovery. Green nodes represent the seed contigs, which have matches to target genome bins in meta-S. The contigs recovered in the increase and decrease steps are shown in orange nodes. Moreover, in the connection step, target contigs are recovered based on path extension, which are also shown in orange nodes. Blue nodes are the target contigs that cannot be recovered by MGA. **(e)** The contig connection graph of *Prevotella salivae*. Each node denotes a contig, and it shares the same scheme as that in Fig. 1d.

But this improvement was obtained at the sacrifice of purity (an average contamination rate of 14.6%), indicating that low initial genome coverage could not provide sufficient information and may introduce contamination. For additional evaluation, we first compared the coverage of these genomes with CONCOCT and metaBAT (ref. 10) using all meta-O and meta-S contigs. As shown in Supplementary Table 2, MGA exhibited much improved performance on capturing highly fragmented genomes compared to CONCOCT and metaBAT. We further compared MGA assembly results with those by metaSPAdes (ref. 26) and IDBA-UD (ref. 27) (Supplementary Table 4). Compared with metaSAPdes, MGA exhibited considerably increased continuity and decreased number of ambiguous bases on six out of the seven genomes. In addition, MGA maintained noticeable increase of assembly continuity

across all genomes compared with IDBA-UD. Similarly, MGA based on metaSPAdes achieved the highest continuity among all the assemblies, indicating an excellent assembly performance of MGA. In metagenomic assembly, contigs of closely related species are tangled together by common genomic region; thus, the assembler tends to misassemble sequences from different species into chimeric contigs[3,5]. To assess the purity of MGA assembly, three assembled species of *Prevotella* were compared with the genomes in the same genus in the oral reference database, and the ANI values were calculated. All of the three genomes showed a high ANI value ($>$97%) within the same species but a low value ($<$77%) compared with other species (Supplementary Table 5), indicating a high specificity of the MGA algorithm for untangling contigs from different organisms.

To investigate the performance of the three key steps (increase, decrease and connection) in MGA, we examined both the sensitivity and specificity rates for each step. As shown in Fig. 2b,c, the increase step achieved relatively high sensitivity (approximately 66.4%) but low precision (approximately 4.2%) at recovering targeted contigs from meta-O using SVM. This result was primarily attributed to the fact that the minimum length for predicting target contigs was set to 300 bp, which did not contain sufficient composition signals for discrimination. However, after the decrease step, the precision increased sharply (approximately 60.2%), with a slight drop of sensitivity. The precision and sensitivity remarkably increased in the connection step to 92.5% and 90.8%, respectively. To obtain a more intuitive and comprehensive understanding of the MGA algorithm, we visualized these steps by extracting a component (Fig. 2d) from the contig connection graph of *Prevotella salivae* (Fig. 2e). The meta-S contigs belonging to the target genome were mapped to the meta-O contig connection graph and used as 'seeds' for recovering the remaining contigs. After the increase and decrease steps, more target contigs were recognized and taken as landmarks on the meta-O contig connection graph to supervise the path-searching algorithm. Through the connection step, target contigs located on the paths between two landmarks were recovered, the layouts of target contigs on the meta-O contig connection graph were determined, and scaffolds were thus assembled. The target contigs (blue nodes) were neglected because they failed to be detected. A similar visualization was also performed in *P. aeruginosa* (Supplementary Fig. 9).

**De novo identification of strain-level variation.** To explore the ability of our approach to reveal strain dynamics, we applied metaSort to a human fecal sample and separated one meta-S subset using FCM (Supplementary Fig. 10). Using the Illumina HiSeq2000, we generated 203 and 118 million paired-end reads for the meta-O and the meta-S samples, respectively. To facilitate the *de novo* variation detection after target genome assembly, SOAPdenovo2 rather than metaSPAdes or IDBA-UD, which merged the variation into consensus sequences, was chosen to assemble the meta-O reads without merging the bubble structures. As a result, the meta-O assembly produced 865,280 contigs with an N50 contig length of 403 bp, and the meta-S assembly contained 261,858 contigs with an N50 length of 2,739 bp. By applying BAF and MGA, we successfully recovered 12 genomes, with genome coverage rates ranging from 83 to 99% and a 2.8- to 29.7-fold NGA75 increase (Supplementary Table 6 and Supplementary Fig. 10). Five out of the 12 species contained an extremely short ($<500$ bp) NGA75 length in the meta-O assembly (Fig. 3a), implying that these species could not be clustered by existing binning methods. However, metaSort successfully recovered their genome sequences in meta-O (average of 90.0%) and remarkably increased the NGA75 length (average of 10.5-fold), exhibiting a substantial advantage over the binning methods (Supplementary Table 6). We also found that the primary factor responsible for these highly fragmented assemblies was the large number of bubbles (Fig. 3b, number of points), which were mainly caused by sequencing errors, repetitive sequences or genetic variants from closely related genomes. The successful recovery and assembly of these species demonstrated the advantages of MGA in not only genome assembly but also *de novo* variation detection by introducing contig connection graphs.

Previous reports have demonstrated the importance of strain-level variation, which may reflect an adaptive dynamic response to environmental changes[28]. With this goal in mind, we extracted

bubbles that were attributed to genetic variants using sequence divergence and sequencing depth. Finally, simple bubbles with at most three walks were utilized to perform strain-level variation detection. Among the 12 recovered species, at least four of them exhibited strain-level variation (Fig. 3b). For example, taxonomic analysis indicated that there were at least two strains in *Bifidobacterium breve*. Moreover, we employed the marker gene-based method MetaPhlAn to identify microbial compositions from metagenomes. Among the nine species detected by MetaPhlAn, only *Eubacterium rectale* contained more than one strain (Supplementary Fig. 11). Such a difference indicates that traditional marker gene-based methods may not be applicable to recognize strain- or subspecies-level variation from metagenomic reads, whereas dissecting the assembly graph in this study provides a novel and efficient means of *de novo* and high-resolution intraspecific diversity detection.

To compare the levels of intraspecific variation among different species, we calculated the alpha diversity of all the bubbles and bubble distances in the assembly graph. The first metric was obtained using the sequencing depth of the walks in a bubble structure. The second metric, bubble distance, utilized the distance between two adjacent bubbles to represent the density of variations. Among the four multi-strain-containing species, both metrics exhibited strong signatures of intraspecific genomic diversity (Fig. 3c). Interestingly, for *Ruminococcus lactaris, Bifidobacterium longum* and *Veillonella* sp., the distribution of bubble alpha diversities revealed a tendency towards enhancement. On the contrary, the bubble distances exhibited a trend of reduction, implying that alpha diversity and bubble distance may explain intraspecific diversity in different aspects. We further employed these two metrics to compare the intraspecific diversity of the shared species between oral and gut samples. As shown in Fig. 3d, compared with the oral sample, *Streptococcus parasanguinis* exhibited reduced diversity ($P=0$, Wilcoxon rank-sum test), but *Veillonella* sp. represented significantly increased diversity in the gut sample ($P=4.1\times10^{-5}$, Wilcoxon rank-sum test), implying that these bacteria may have undergone different selective pressures in human oral cavity and gut environments.

**MetaSort unveils a novel epibiont community on marine kelp.** We next conducted shotgun metagenomic sequencing on a kelp metagenomic sample to further explore the ability of metaSort to recover species from novel microbial communities with limited reference genomes. Overall, 59 million read pairs were generated and assembled, yielding 218 Mbp across 150,059 contigs (N50 length of 20,201 bp). The application of FCM to this sample produced three subsets of cells, named P2, P3, P5 and the number of cells was $2.4\times10^4$, $3.0\times10^4$ and $7.4\times10^4$, respectively (Supplementary Fig. 12a). Taxonomic annotation results revealed that these subsets were dominated by distinct genera but exhibited reduced taxonomic diversity compared with the original sample (Supplementary Fig. 12b), indicating the high efficiency of FCM in reducing microbial complexity. Assembly of the meta-S data sets yielded 42,055 scaffolds with an N50 length of 5,737 bp. By applying the BAF and MGA algorithms, we successfully reconstructed 75 genomes that were enriched in meta-S and meta-O, with N50 lengths varying from 1.9 Kbp to 2.3 Mbp (Fig. 4a). Phylogenetic analysis demonstrated that these reconstructed genomes belonged to the phyla *Proteobacteria*, *Planctomycetes*, *Actinobacteria*, *Bacteroidetes* and *Cyanobacteria*. It is worth noting that although the genome sequence is the foundation for metabolic and evolutionary studies of an organism, only 35 genomes belonging to the phylum *Planctomycetes*

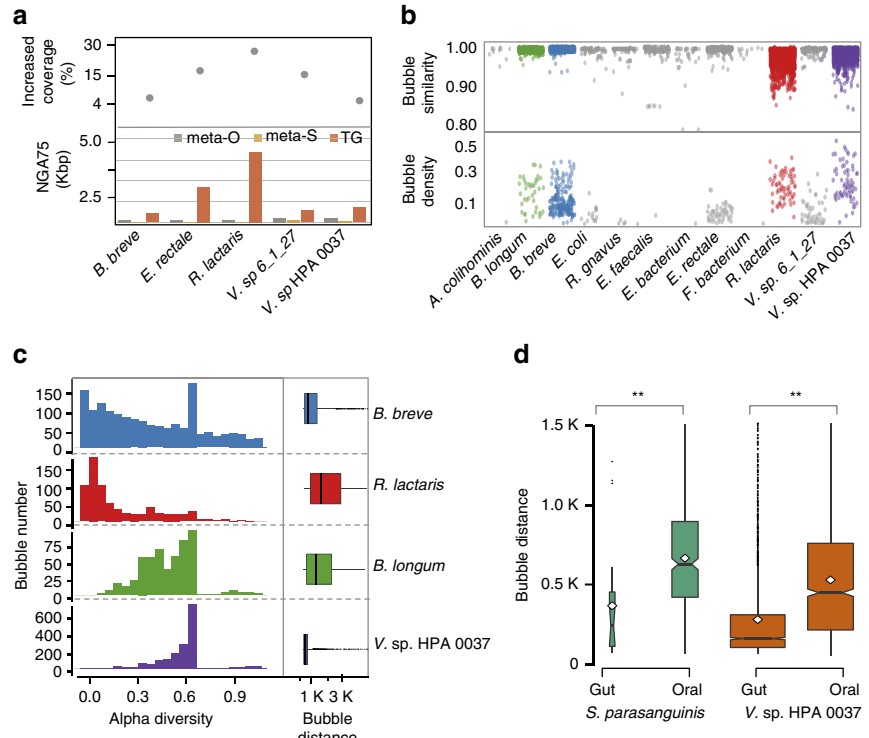

**Figure 3 | Strain-level variation in the gut metagenomic sample.** (**a**) The NGA75 length (bar plot) and increased genome coverage (point plot) of the top five species (*Bifidobacterium breve*, *Eubacterium rectale*, *Ruminococcus lactaris*, *Veillonella*. sp. HPA 0037 and *Veillonella*. sp. 6_1_27) in the gut metagenome, which contain the largest number of components after MGA assembly. (**b**) Bubble density and similarity of each component in the contig connection graph of each species. The four species containing stain-level variations are coloured in green, blue, red and purple, respectively. (**c**) Comparison of the levels of intraspecific variation among the four species using the alpha diversity and bubble distance. The distribution of bubble distance is illustrated by box plot and the alpha diversity is shown by bar plot. (**d**) Comparison of bubble distance of *S. parasanguinis* and *V*. sp. HPA0037 in oral and gut sample, respectively. Significant differences are observed between these two samples (**$P < 0.01$, Wilcoxon rank-sum test).

are currently available in the NCBI database. In this study, we successfully assembled 16 new genomes in this phylum using three mini-metagenomes from one FCM separation, which strongly demonstrates the ability of metaSort to reconstruct novel genomes from less-explored microbial communities.

To evaluate the completeness and possible contaminations in the assembled genomes, we applied CheckM (ref. 29) to these assembled genomes (Supplementary Table 7). Based on this method, the genome completeness was estimated to be an average of 78%. Among them, five bacterial genomes were estimated to be 100% complete. In contrast, recent single-cell sequencing-based studies of free-living aquatic bacteria obtained an average completeness of 40 to 55% (refs 30,31). In addition, both sequencing depth and nucleotide composition analyses revealed a high level of homogeneity of these assemblies, further demonstrating that there should be no large-scale contaminations or chimeras. For example, a sequencing depth analysis of the top five largest scaffolds in HD560 showed that the depth of most bases fluctuated over a small range and followed a normal distribution ($P = 8.92$, Kolmogorov-Smirnov test) (Supplementary Fig. 13a,b). A pairwise comparison of these scaffolds using the tetra-nucleotide Z-statistic correction coefficients[32] also indicated that the sequencing composition among these scaffolds were highly related (Supplementary Fig. 13c). The contamination of the assemblies was estimated to be approximately 3.4% on average based on the CheckM analysis. To better assess metaSort, we further applied metaBAT and CONCOCT on the assemblies from both meta-O and meta-S for

comparison. As shown in Supplementary Fig. 14, metaBAT recovered 32 genomes, with contamination rate varying from 0 to 599% (an average of 35%). By using CONCOCT, we obtained 35 genomes with contamination rate varying from 0.1 to 200% (an average of 17%). Moreover, metaBAT and CONCOCT exhibited considerably decreased contig N50 length compared with that of metaSort, indicating that MGA algorithm greatly improves genome recovery and assembly.

We further validated the accuracy of the assembled genomes using long PacBio reads[33]. Filtering the reads that had low nucleotide identity and short mapping lengths, 106,005 reads remained and were further aligned to the 75 assembled genomes using BLASR (ref. 34). As shown in Fig. 4b, 90% of PacBio reads continuously located to one scaffold with high nucleotide identity, whereas only a very small proportion of them (4%) were mapped to multiple genomes. Furthermore, 5% of PacBio reads located to more than one scaffold that belonged to the same genome. Regarding the 4% multiple mapped reads, the average alignment length was approximately 2 Kbp, and most of them could be mapped to more than three genomes (Fig. 4d,e), suggesting these reads were likely to be repetitive sequences shared by different genomes. These results collectively suggested the high accuracy of the MGA assembly algorithm.

**Assembled genomes shed light on kelp-bacteria interactions.** To determine how the functional specificity of the kelp microbiome developed in an epiphytic manner, we compared the

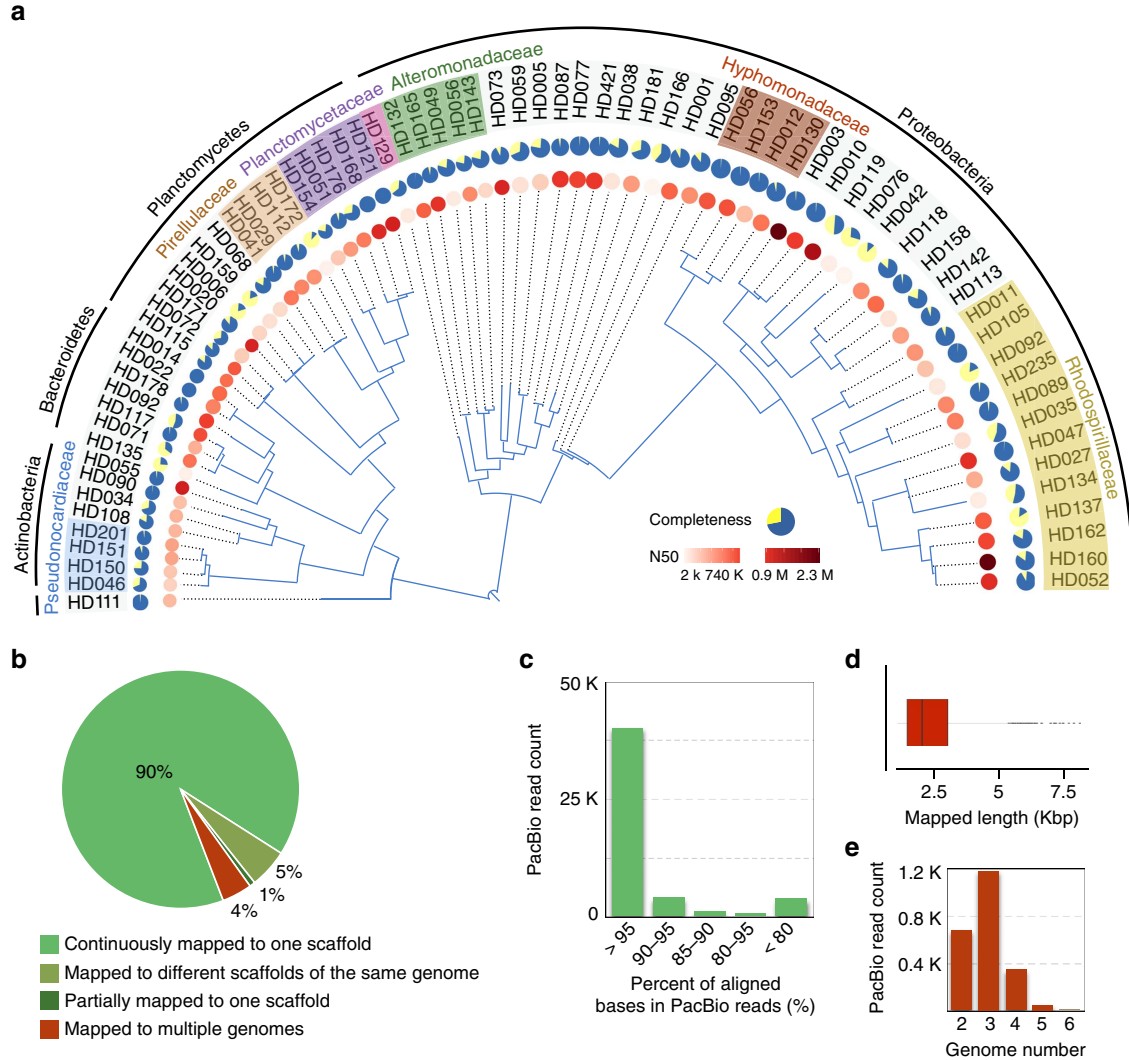

**Figure 4 | Phylogenetic tree and assembly validation of the reconstructed kelp bacterial genomes. (a)** Phylogenetic tree of 75 assembled kelp bacterial genomes belonging to five phyla. Genomes assigned to seven major families are highlighted in colour, whereas the remaining genomes cannot be assigned to the family level based on the taxonomic classification of PhyloPhlAn. The estimated completeness of each genome is represented by pie plot and the N50 length is illustrated by heat map. **(b–e)** Validation of the assembled kelp bacterial genomes using long PacBio reads. PacBio reads are mapped to the assembled genomes using BLASR. After the mapped reads are filtered by nucleotide identity and alignment length, the remaining reads are divided into four categories according to the number of hits **(b)**. The percent of aligned bases in the PacBio reads that are uniquely mapped to one scaffold **(c)**. Regarding the 4% multiple mapped reads, the average mapped length is short **(d)** and these reads tend to be found in multiple genomes **(e)**.

KEGG pathways of the recovered genomes with two other typical ecosystems, the human gut and the ocean. Significant differences in enriched functional categories were identified among the three ecosystems (Supplementary Table 8). Most notably, those for energy metabolism, key nutrient supplements, attachment mechanisms and surface competition were considerably enriched ($P < 1 \times 1^{-10}$, Fisher's exact test, Bonferroni corrected) in the kelp microbiome (Fig. 5a). This result demonstrated a complementary and mutualistic relationship between kelp and its epiphytic bacteria, where bacteria may benefit from the ready availability of polysaccharides produced by the host alga. The epiphytic bacteria subsequently enhance the growth and nutrient uptake of the host alga by producing bioactive substances, including vitamins[35] and hormones.

Because horizontal gene transfer (HGT) events typically involve changes that may endow bacteria with new functions and thus give rise to adaptation to the environment[36], we identified HGTs in the recovered genomes in the kelp microbiome using both GC content deviation and phylogenetic analyses (Supplementary Table 9) and further assigned functions by scanning the EGGNOG (ref. 37) database. As expected, the enriched gene functions ($P < 0.01$, hypergeometric test, Bonferroni corrected) of HGT events were mainly symbiosis-associated, such as polysaccharide degradation, succinoglycan biosynthesis[38] (a symbiotically important exopolysaccharide) and nitrogen fixation (Fig. 5b and Supplementary Table 10). This finding may indicate that strong selective pressure may shape these microorganisms to sustain their host and benefit from the symbiotic relationship.

Polysaccharide degradation is essential for various bacterial communities, which take up these biopolymers as key nutrient source[39,40]. To assess the potential of polysaccharide degradation in the kelp microbiome, we annotated the carbohydrate active enzyme (CAZyme) spectrum by searching against the CAZy database[41]. This analysis identified 12,054 (3.4% of total proteins) putative CAZymes belonging to 208 CAZY families, suggesting

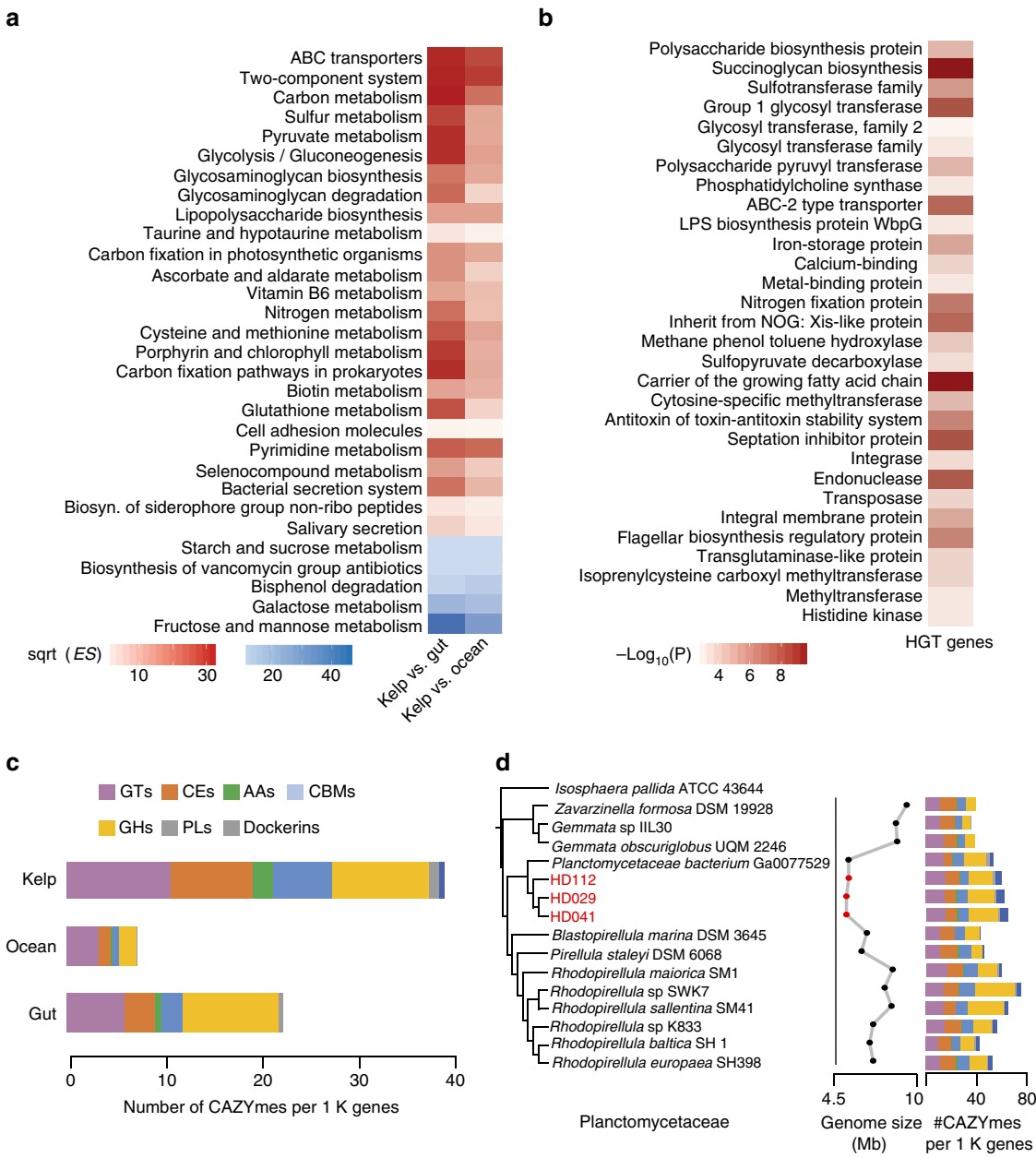

**Figure 5 | Functional insights into the assembled kelp bacterial genomes.** (**a**) Comparison of the KEGG pathways of the assembled kelp bacterial genomes with the ocean and human gut microbiomes. Heat map shows the enrichment score between kelp and gut (left), kelp and ocean (right). $P < 1 \times 1^{-10}$, Fisher's exact test and Bonferroni corrected. Red colour represents enriched functions in the assembled kelp bacterial genomes and blue colour shows enriched functions in the gut or ocean microbiomes. (**b**) The enriched EGGNOG functions of horizontally transferred genes. Heat map shows the $-\log_{10}$ of $P$ value (hypergeometric test). (**c**) Comparison of the number of CAZyme genes among the three microbiomes. (**d**) Features of the three kelp bacterial genomes (HD112, HD029 and HD041) assigned to the Planctomycetaceae family. Genome size is shown in curved line, and the number of CAZyme genes in each genome is shown in bar plot.

the vast diversity and potential of polysaccharide degradation in these epiphytic bacteria. Among them, GTs (29.4%) and GHs (24.6%) were the most abundant classes, followed by CEs, CBMs, AAs, PLs, dockerin and cohesin. In particular, seven alginate lyase families (PL6, PL7, PL9, PL10, PL14, PL17 and PL18) for degrading alginate, which is the main component of the cell wall in brown algae[35], and four laminarinase families (GH16, GH17, GH64 and GH128) for degrading laminarin, which is a unique storage polysaccharide in kelp[35], were observed. Notably, the kelp microbiome possessed many more CAZymes (38 per 1,000 genes) than the other two

ecosystems (Fig. 5c) (7 and 22 in ocean and human gut, respectively). In addition, a large number of exocellular carbohydrate-degrading complexes called cellulosomes[42] that consist of cohesins, dockerins and surface layer homology modules were detected (Supplementary Fig. 15a). By playing a major role in carbon turnover, cellulosomes enable bacteria to degrade plant cell walls and to convert polysaccharide compounds in a highly efficient manner[42]. An analysis of kelp microbial proteins with hits to cellulosome-associated modules revealed that 1.1% of the total CAZymes were dockerins. Compared with the other two microbiomes, the kelp

microbiome exhibited prominent enrichment in cohesin and dockerin modules.

As an evolutionarily deep-branching phylum in the domain Bacteria, *Planctomycetes* exhibit unusual properties, including non-typical peptidoglycan cell walls and the appearance of internal compartments[43–45]. In this study, we successfully discovered 16 nearly complete *Planctomycetes* genomes from the kelp epiphytic community, which provide invaluable new resources for understanding the metabolic potential and genome evolution of this phylum. As shown in Fig. 5d, comparative numbers of CAZyme genes were detected in the genomes of *Rhodopirellula baltica* SH1 and *Rhodopirellula*. sp SWK7, which live on the surface of macroalgae and use polysaccharides as their main energy source[46]. However, the number of dockerins in kelp bacteria was considerably increased compared with other related organisms in this phylum, and a phylogenetic analysis revealed that the dockerin genes in HD041 and its close relatives (HD112 and HD029) are likely derived from gene duplication (Supplementary Fig. 16). Algal cell walls contain plentiful sulfated carbohydrates, such as ulvan and fucoidan[35,40,47]. An important feature of species in genus *Rhodopirellula* is that they possess a large number of sulfatase genes, which are essential in the degradation of sulfated polysaccharides[46]. The closely related kelp bacterial genomes, however, contain many fewer sulfatase genes than *Rhodopirellula* (Supplementary Fig. 15b), which may be partly because their host *Saccharina* contains fewer sulfated polysaccharides. Interestingly, the three new species identified in this study had the smallest genome size (approximately 4.9 Mb) in the Planctomycetaceae family, representing a genome reduction compared with other related organisms (6.2–10.1 Mb). A functional enrichment analysis revealed that the lost genes compared with *Rhodopirellula* spp. were mainly involved in the biosynthesis of amino acids and carbon metabolism (Supplementary Fig. 17).

## Discussion

This study presents a novel experimental and bioinformatic framework, metaSort, for the effective construction of bacterial genomes from metagenomic samples. The main advantage of metaSort is that it provides a sorted mini-metagenomic approach based on FCM and single-cell methodologies. To combine the sorted mini-metagenome (meta-S) and the original metagenome (meta-O), we developed two new computational algorithms, BAF and MGA, with the aim of reconstructing genomes from small subsets of cells and variation calling. In addition, to improve the applicability of metaSort, these two algorithms are designed as stand-alone software packages and can serve as a hot-plug scaffolder for current *de Bruijn* graph based metagenomic assemblers, such as metaSPAdes and IDBA-UD. This approach greatly accelerates our ability to capture and assemble high-quality genomes from various environmental samples, which will undoubtedly benefit the field of metagenomics.

With the development of new sequencing technologies, high-quality genomes can be obtained using PacBio and Illumina TruSeq Synthetic (Moleculo) long reads[48], which makes metagenome assembly considerably easier. However, the high error rate of PacBio long reads makes it difficult to distinguish authentic variations from sequencing errors in metagenomes. For example, Kuleshov *et al.* showed that the assembly of PacBio reads had a five-fold higher indel rate and only 88% of single nucleotide variant calls could be confirmed by Illumina short reads[48]. Although Moleculo overcomes this drawback and has been applied to assemble bacterial genomes and to identify substrains in microbiomes[48,49], both long-read-based

technologies are greatly influenced by the unevenness of species richness in the microbial community. High-abundance bacteria have increased probability to be sequenced compared with moderate- or low-abundance species, thus resulting in an overwhelming volume of 'useless' data or preventing the sub-assembly of the Moleculo long reads. Howe *et al.* surveyed the HGMC data set and found that 60% of the total reads represent only 2% of captured bacteria, which could be discarded with no effect on assembly[50]. Kuleshov *et al.* applied both Moleculo long-read and short-read sequencing technologies to a gut metagenome and identified 178 species, among which only 51 (28.7%) with abundance <5% were uncovered by Moleculo long reads[48]. In contrast, metaSort uses FCM to generate meta-S, which can effectively avoid the interference of highly abundant bacteria by sorting cells by size. In addition, users can control the number of cells in each meta-S using FCM. For example, if only one cell is separated, it is a typical application of single-cell sequencing. In contrast, the taxonomic profile of meta-S will be similar to that of meta-O if a large fraction of cells are selected. Therefore, metaSort provides a manageable and dynamic means of generating meta-S and can reduce the complexity of metagenomic by controlling the number of separated cells. Moreover, compared with traditional single-cell sequencing, these subsets of cells with few overlaps can significantly improve the throughput and thereby provide a more economical means of capturing diverse genomes. Moreover, other sorting methods, such as specific nucleotide acid probes and magnetically labelled antibodies, can be applied to separate target bacteria, which will greatly expand the application of this method.

A crucial feature of metaSort is the binning and reconstruction of sequences of the target genome based on the combination of meta-S and meta-O. Recently developed unsupervised binning methods face challenges either in the large number of samples or the short length of the assembled contigs. These challenges greatly limit the application of such methods to complex metagenomic samples for the following reasons: (i) for co-occurrence-based binning methods, assembling a large number of metagenomes may cause a computational bottleneck in most computers[51], and (ii) genomes that have intra-species variations but low abundance are generally assembled into small fragments, which are not applicable to most binning methods. MetaSort, however, outperforms these methods based on the following: (i) it does not require a large number of samples for constructing co-occurrence profiles and thus will significantly reduce the burden of computing; (ii) it employs meta-S to reduce the complexity of assembly and to recover low-abundance species; and (iii) it takes advantage of the meta-O contig connection graph to recover short target contigs, which are overlooked by most current binning approaches. Moreover, by traversing the target genome contig connection graph, the order of target contigs can be easily determined, and the sequences of these ordered contigs can be merged into long scaffolds. An improved metagenome assembly will greatly benefit downstream analyses, for example, gene prediction and horizontally transferred gene detection. It should be noted that FCM-based cell separation is not a prerequisite for the utility of metaSort. Users can use closely related genomes to replace meta-S and then run BAF and MGA to recover target contigs from meta-O. Furthermore, if no meta-S data set is provided, the computational tools implemented in metaSort will perform *de novo* binning of the assembled contigs in meta-O, and the results will then be processed by MGA.

Existing variation detection algorithms often depend on the availability of reference genomes[24,52]. When reference genomes are lacking, variation detection is typically performed

in a post-assembly manner. However, traditional metagenome assemblers often produce highly fragmented contigs or merge different variants into consensus sequences[3]. Therefore, most genomic variants in the metagenomic assembly cannot be detected. A recent effort to use variation-aware contig graphs to detect variation relied heavily on manual inspection, and it is not applicable to detect strain-level variations[53]. In contrast, metaSort can recognize variation from the contig connection graph during the assembly process, which provides a high-resolution map of genetic variation in metagenomes.

It is becoming clear that microbes colonizing the surface of macroalgae are essential for the growth and development of the host[40]. To comprehensively understand the genetic basis of macroalgae-bacteria interactions, obtaining genomes of representative bacteria is a crucial step. However, few reference genomes are available, which is partly due to the difficulty in cultivating these genomes. Our assembly of 75 bacterial genomes using metaSort remarkably illuminates kelp-bacteria interactions. Compared with ocean and gut microbiomes, the kelp microbiome is rich in energy metabolism, key nutrient supplements, attachment mechanisms and surface competition. In the assembled genomes, we identified 12,054 putative CAZymes belonging to 208 CAZY families, indicating a vast diversity and potential for polysaccharide degradation in these epiphytic bacteria. Notably, the kelp microbiome possesses many more CAZymes (38 per 1,000 genes) than the other two ecosystems (7 and 22 per 1,000 genes, respectively), indicating that these epiphytic bacteria have adapted to their host environment. Interestingly, HGTs may contribute to the expansion of CAZyme-related genes in these bacteria. The high-quality genomes generated in this study will provide an opportunity to study the roles of epiphytic bacteria in the morphological development, growth, defense and nutrient uptake of their algal hosts.

## Methods

**Meta-O contig connection graph construction.** SOAPdenovo v2.04 was used to assemble metagenomic reads with following parameters: all -K 41 −m 71 −u −R. The resulting contigs were used to construct the meta-O contig connection graph. First, we refer a directed string graph as contig connection graph $G = (V, E)$ that has a set of nodes $V = \{v_1, v_2, \ldots, v_n\}$ and a set of edges $E = \{(v_1, v_5), (v_1, v_3), \ldots, (v_m, v_k) | v_m, v_k \in V\}$ that are formed by pairs of overlapping contigs[54]. The set of nodes is created by assigning each contig $c \in C$ to a unique node. The graph $G_1 = (V_1, E_1)$ is a subgraph of $G = (V, E)$ if (i) $V_1 \subseteq V$ and (ii) every edge of $G_1$ is also an edge of $G$. The subgraph $G_1 = (V_1, E_1)$ is a component of $G = (V, E)$ if $G_1$ satisfies three conditions: (i) $V_1 \subseteq V$; (ii) every edge of $G_1$ is also an edge of $G$; and (iii) any two nodes in $G_1$ are connected to each other by a path, and no paths can be found to connect the nodes between $V_1$ and $(V - V_1)$. The meta-O contig connection graph was bi-directed because both contigs were derived from the forward strand and the reverse strand of DNA sequences. Then, paired-end read-mapped links (Plinks), which are defined as distant connections between two different contigs that are supported by a number of paired-end reads, are created. To feasibly represent the distance between two contigs that are supported by Plinks for scaffolding, the insert length of the sequencing library is estimated using normally mapped paired-end reads based on a mixed Gaussian distribution and EM algorithm.

**Meta-O CCG partition.** The meta-O contig connection graph is partitioned into many small subgraphs such that each subgraph represents a partial sequence of a species. This partition is based on the assumption that genomes from different species seldom contain a common sequence, as previously described by Peng et al. This partitioning is achieved by a greedy algorithm, which repeatedly checks all outward edges of each node with out-degrees larger than 2. If all the paths with length $\leq w$ (for example, 400 bp) that start from the outward edges of a node $u$ cannot converge into another node $v$, all the outward edges of vertex $u$ are removed from the meta-O contig connection graph.

**Meta-S contig binning.** Mini-metagenomic reads were assembled using SPAdes v3.15 (with SC and careful mode). To improve the binning efficiency, a new algorithm (binning and fishing, BAF) was developed to bin the meta-S contigs by introducing connected contigs from meta-O (Supplementary Fig. 1). BAF starts by clustering the meta-S contigs into bins based on TNF. Meta-S bins are then mapped to meta-O contigs using NUCMER (ref. 55) to locate meta-S bins on the meta-O contig connection graph. Links are created if two meta-S bins on the meta-O contig connection graph satisfy one of two criteria: (1) both meta-S bins have an end-to-end overlap (at least 40 bp) with a node; or (2) there is only one path between these two meta-S bins, and the nodes on the path cannot be aligned to any other meta-S bins except themselves. Subsequently, the meta-S bins and the paths linking the meta-S bins are extracted to form a small graph. Finally, the small graph is partitioned into many components based on graph connectivity, and the meta-S bins in the same component are clustered together into merged bins. The merged bins and the remainder of the meta-S bins that do not participate in the process of merging constitute the target genome bins.

**Machine learning and graph-based algorithm (MGA).** After BAF is performed, a set of target genome bins are generated. Each bin represents a partial genome assembly, which may cover 10-90% of the complete genome (see Fig. 2). The MGA algorithm was developed to recover the remaining contigs of the target genome and then create a combined assembly. MGA can be divided into the three steps described below.

(1) The increase step. Similar to the method used in PhyloPythia[21], for each target genome bin, a supervised SVM-based algorithm is used to recover the remaining target genome contigs from meta-O based on the positive, negative and test data sets. MGA takes the sequences in the target genome bins as the positive data set. Subsequently, negative and candidate data sets are prepared using the Spearman Footrule distance metric[20], which can quantify the difference between two TNF distributions from their fragments. The intra-species distance distribution is obtained by calculating all the pairwise distances of the sequences in the target genome bins. The distances of all the meta-O contigs are recorded by comparing these values with the sequences in the target genome bins, and those meta-O contigs with distances twice as large as the standard deviation of inter-species distance are classified into the negative data set. Otherwise, the sequences will be classified into test data set. Finally, SVM trains the classification model based on the positive and negative data sets using codon usage as a feature vector and subsequently candidate target genome contigs are predicted.

(2) The decrease step. With candidate target genome contigs predicted by SVM, the sequencing depth of target genome and the partitioned subgraphs of the meta-O contig connection graph are employed to filter false-positive contigs. Because the sequences in the target genome bins are generated by MDA technology, their sequencing depth is highly uneven and does not represent the true sequencing depth of the target genome. MGA first maps the meta-O contigs to the sequences in the target genome bin to obtain seed contigs, which can feasibly represent the target genome. Next, MGA utilizes a Gaussian mixture model fit with an expectation-maximization algorithm to estimate the sequencing depth distribution of the seed contigs. Candidate target genome contigs with sequencing depths outside the range of two standard deviations of the depth distribution are discarded. After filtration, the remaining target genome contigs and seed contigs, referred to as landmarks, will be used for traversing the meta-O contig connection graph in the next step.

(3) The connection step. Rather than traversing the entire meta-O contig connection graph, MGA uses a breadth-first searching algorithm, which starts at each landmark and searches against the contig connection graph using a given step size to identify the path that can connect two landmarks. Once all landmarks have been visited, the landmarks along the search path are extracted to build a target genome contig graph. Prior to generating the final scaffold, MGA executes two functions, 'trimming tips' and 'merging bubbles', to simplify the target genome contig graph. A 'tip', which is defined as a chain of nodes disconnected on one end, is removed. A bubble structure is defined as several similar paths with the same start node and end node in the contig graph. We further defined nodes that are shared by at least two paths in a bubble structure as uniform nodes; otherwise, nodes are termed divergent nodes. Let $v$ be a node in the target genome contig graph with branches (the out degree of $v > 1$). Following each branch, we search outward from $v$ for a set of walks, $W$, that satisfies the following two conditions: (i) all walks end at a common node $u$, and (ii) no node included in any walk between $v$ and $u$ can connect a node that does not belong to $W$. Once a set of walks that fulfils these criteria has been found, the sequences of the divergent nodes among the walks are compared using a Dynamic Programming algorithm to discard the bubbles caused by repetitive elements. If the sequence identity is higher than a predefined threshold (by default, 95%) in all cases, one walk containing the smallest depth variation with target genome is retained, and the others are recorded and removed from the graph. Finally, the simplified target genome contig graph is assembled into scaffolds. Information on the removed walks is retained for downstream analyses.

**Two new metrics to evaluate metagenomic variation.** In the contig connection graph of a metagenome, bubbles may represent sequencing errors, repetitive sequences or genetic variants from closely related genomes. In the connection step of MGA, we filter the bubbles caused by repetitive sequences using sequence identity. Because sequencing errors may cause errors in the detected variations, we further filter the bubbles by removing walks with a sequencing depth $< 2$-fold from the bubble structure. Moreover, we propose the following two metrics to qualify the

variations:

$$\text{bubble density} = \frac{\sum_{i=1}^{m} LD_i}{2 \times \left( \sum_{i=1}^{m} LD_i + \sum_{i=1}^{n} LU_i \right)}$$

$$\text{bubble identity} = 1 - \frac{\sum_{i=1}^{m} Mis_i}{\left( \sum_{i=1}^{m} LD_i \right)/2 + \sum_{i=1}^{n} LU_i}$$

where LD is the length of a divergent node, LU is the length of a uniform node, $n$ is the number of uniform nodes on the component, and Mis is the number of mismatches among the divergent nodes. The bubble distance, which is defined as the sequence length between the two nearest divergent bubbles, is calculated using an algorithm that iteratively deletes the bubble nodes in a component and calculates the accumulated contig length between two pendant ends (nodes with in-degree = 1 and out-degree = 0).

**Strain-level variation identification.** We identified strain-level variation-containing species using a logistic regression model. The probability of a species containing strain-level variation is defined as: $svp(x) = 1/(1 - e^{\theta^T X})$, where $X$ is a feature vector containing two values, $m$ and $k$. $m$ is the average component identity where the outliers ($\pm 3$ s.d.) have been removed, and $k = \log(m/n)$, where $n$ is the component number of the target genome contig graph. $\theta$ is the weight of every feature in $X$. To obtain the $\theta$ value, we used the 100 genomes in the mock data set. Then, we defined a regression model as follows:

$$- \sum_{i=1}^{100} y_i \log(svp(x_i)) + (1 - y_i)\log(1 - svp(x_i))$$

where $y$ equals 1 if the genome contains strain-level variation; otherwise, $y$ equals 0. We applied the gradient descent algorithm to tune the parameters and to obtain the $\theta$ value.

To calculate the abundance of each variant in the bubbles, we first mapped the metagenomic reads to the target genome contig graph to obtain the contig depth. Then, we identified the contributions of individual strains when contigs were shared by multiple strains and solved a weighted non-negative least squares problem for each bubble structure. Due to the depth bias of short contigs, we defined the weight function to represent the confidence level of depth with respect to contig length as follows: $w = 1 - e^{-x \times 0.5/100}$, where $x$ is the length of the contig.

We further introduced alpha diversity, a metric of the number of strains and the proportion at which each strain is represented in the species, to compare the levels of intraspecific variation among different species. The alpha diversity in each bubble was calculated as follows:

$$H = - \sum_{i=1}^{n} Pi \times ln(Pi)$$

where Pi is the abundance ratio of each variant in the bubble, and $n$ is the variant number.

**Microbial sample collection and preparation.** Approximately 10 ml of saliva was collected from a healthy human volunteer who was free of systemic diseases and other oral diseases, without prosthetic dental appliances, had never received periodontal therapy and had not taken any antibiotics in the past three months[56–58]. The sample was centrifuged at 2,500$g$ for 30 s at room temperature to remove large particles. The supernatant was stored in sterile plastic tubes and frozen at $-80\,°C$ for further processing. Approximately 5 g of a fecal sample from a newborn was collected in a 50-ml sterile tube, suspended in sterile $1 \times$ phosphate-buffered saline and vortexed for 5 min. Then, the sample was centrifuged at $2,500 \times g$ for 30 s at room temperature to remove large particles. The supernatant was transferred to sterile plastic tubes and frozen at $-80\,°C$ for storage.

Female gametophytes of kelp[59] (*Saccharina japonica*, SJ) were cultured at $10 \pm 1\,°C$ and 5 μmol photons m$^{-2}$ s$^{-1}$ with a 12:12 h light/dark photoperiod. Microorganisms colonizing the surface of SJ were harvested using a shaker (Thermo Fisher Scientific, Waltham, USA). Then, bacterial cells were collected in sterile plastic tubes and stored at $-80\,°C$ for further processing.

**Cell sorting.** Fresh cells from the above samples were centrifuged at 16,000$g$ (15 min) and re-suspended in sterile $1 \times$ phosphate-buffered saline at $10^6$ to $10^7$ cells ml$^{-1}$. The cell suspensions were filtered through BD Falcon Cell-Strainer Caps (352340) and then sorted on a BD Influx flow cytometer (BD Biosciences, USA) using a 488-nm argon laser for excitation and 70 μM nozzle orifice filter cartridges with a sheath pressure below 40 psi. For optimal sorting performance, the cell sorting accuracy was tested prior to library sorting using a mixture of polystyrene fluorescent beads (530 nm, 950 nm, 1.6 μm, 2.0 μm and 3.2 μm, respectively), which was purchased from Nano-Micro Co., Ltd (Suzhou, China). Then, the sorted beads were analysed by FCM for purity of which greater than 98% was acceptable using a high purity sorting mode. Subsequently, sort gates were specified according to bacterial cells of different sizes in the library, which were characterized by analysing their forward scatter (FSC) and side scatter (SSC) and the number of cells in each gating window on the FSC and SSC plot were

determined. Finally, the sorting gates containing target cells were segregated in 5-ml round-bottom tubes (BD Biosciences, USA) and stored at $-80\,°C$ until further processing.

**DNA extraction and whole-genome amplification.** Cells of specimens sorted by flow cytometry were transferred to 1.5-ml sterile tubes (Eppendorf Ltd., Germany) and centrifuged at 16,000$g$ (15 min) for enrichment. Then, these cells were resuspended in 5 μl of lysis buffer (0.13 M KOH, 3.3 mM EDTA pH 8.0, and 27.7 mM DTT) and incubated at $65\,°C$ for 30 min, after which 5 μl of neutralization buffer (0.13 M HCl, 0.42 M Tris-HCl pH 7.0, and 0.18 M Tris-HCl pH 8.0) was added to stop the reaction. To obtain sufficient DNA for next-generation sequencing library preparation, genomic DNA released from the lysed cells was amplified with the Phi29 DNA polymerase supplied in the REPLI-g Mini Kit (Qiagen, Hilden, Germany). All the components were added on ice (in the order listed in the REPLI-g Mini Kit protocol), mixed and centrifuged gently. After 8-h amplification at $30\,°C$, the Phi29 DNA polymerase was denatured at $65\,°C$ for 5 min. Then, the DNA solution was cooled on ice for 3 min and purified with the QIAGEN Genomic DNA Kit (Qiagen, Hilden, Germany). The quantity of purified DNA was measured using a NanoDrop 2000 spectrophotometer (Thermo Fisher Scientific, Waltham, MA, USA) and the quality was checked by agarose gel electrophoresis. The DNA solution was stored at $-20\,°C$ for further processing.

**High throughput sequencing.** The purified metagenomic DNA was amplified from the sorted cells using a whole-genome amplification kit, and the original metagenomic DNA was fragmented into approximately 180-bp fragments by sonication using Covaris s220 (Covaris). Next-generation sequencing libraries were constructed according to the manufacturer's protocol (Illumina, Inc.), quantified using a Stratagene Mx3000P Real-time PCR Cycler (Agilent, Santa Clara, CA, USA), subjected to cluster generation in a c-Bot automated sequencing system (Illumina, Inc.), and finally sequenced with $2 \times 100$ bp paired-end reads on an Illumina HiSeq 2000 instrument at the Beijing Institutes of Life Science, Chinese Academy of Sciences (CAS). For the PacBio library construction, genomic DNA was sheared to 8 kb using an ultrasonicator and was converted to the proprietary SMRTbell library format using an RS DNA Template Preparation Kit. SMRTbell templates were subjected to standard SMRT sequencing on the PacBio RS system according to the manufacturer's protocol. The raw sequencing reads were trimmed using Trimmomatic[60].

**Synthetic data set.** A data set containing 100 bacterial genomes was simulated with abundance following a power law; the sequencing depth ranged from 5- to 128-fold. Simulated sequencing errors were randomly distributed. The error rate was set at 2%, and the average insert length was 200 bp. The data set was designed to test the ability of MGA to resolve different levels of taxonomic resolution, including strains of the same species. These genomes were divided into three major testing categories according to different taxonomic levels: (i) species: bacterial genomes from the same species but different strains; (ii) genus: bacterial genomes from the same genus but different species; and (iii) $\geq$ family (see SI Appendix, Fig. 3 for details). Calculations of ANI values between two genomes according to BLAST (ANIb) were performed using JSpecies[61].

**Evaluation of the assemblies.** We evaluated the performance of the assembled genomes in oral and gut microbiomes using QUAST because most human oral and gut bacteria have been previously sequenced. For the analysis, we considered the following quality metrics: total assembly length, average scaffold length, maximum scaffold length and NGA75 size.

**Computational and experimental validation of assembled genomes.** The completeness and contamination of the assembled kelp bacterial genomes were evaluated using CheckM. PacBio long reads were mapped to the assembled scaffolds using BLASR to validate the assembly. We first calculated the error rate of the PacBio reads using 100% mapped PacBio reads with a minimum read length of 5 Kbp and the ANI value (97%) between two strains from same species. To address the problem of genomic regions shared by different genomes, we first built feasible mapping regions (FMRs) using PacBio reads larger than 1 Kbp with a minimum read coverage of 80%. Then, the reads that were mapped on the scaffolds were analysed. If the reads were partially mapped to the FMRs, they were considered false mapping and were thus abandoned. If the reads were mapped outside the FMRs, they were accepted if the following two criteria were satisfied: (i) a minimum mapping length of 2 Kbp and (ii) no mapping hits to other scaffolds. Reads mapping to different genomes were required to have the following: (i) at least 1 Kbp mapped length on each scaffold; (ii) an accumulated read coverage greater than 50%; and (iii) a mapping region located outside the FMRs of the target scaffold. After the mapped PacBio reads were classified, the number of PacBio reads in each catalogue was calculated and normalized to the total number of mapped PacBio reads. In addition, the nucleotide identity of continuously mapped PacBio reads was classified into five levels (80–100%), and the number of mapped PacBio reads in each level was counted. For PacBio reads that were mapped to

multiple genomes, the alignment length and number of the mapped genomes were also calculated.

**Functional annotation.** We performed gene prediction using PROKKA (ref. 62) v1.11 for the assembled genomes, and the predicted proteins were input into PhyloPhlAn (ref. 63) v0.99 to assign microbial phylogeny and putative taxonomy to each genome. Proteins were assigned to the KO using the KEGG Automatic Annotation Server[64]. To identify CAZymes in the recovered kelp bacterial genomes, we performed CAZyme screening in these assembled genomes and the gene catalogues of the gut and ocean microbiomes (http://www.genome.jp/mgenes). All of the putative proteins were searched against entries in the CAZy database using the dbCAN Web server[65], in which HMMer (ref. 66) was used to query a collection of custom-made hidden Markov model profiles that were constructed for each CAZy family.

**Enrichment analysis.** We counted the numbers of KEGG Orthologs (KOs) in the assembled kelp bacterial genomes and the gut and ocean reference gene catalogues. Functional enrichment was determined using Fisher's exact test and was adjusted for the FDR. KOs with a $q$-value less than 0.01 were considered enriched; these KOs were mapped to KEGG metabolic pathways using publicly available data for each KO in the KEGG website to calculate the pathway enrichment. An enrichment score was proposed for this study and was calculated for pathways that contained enriched KOs. The enrichment score was defined as follows:

$$ES = \frac{T \times \sum_{i=1}^{T}(-\log 10(P))}{N},$$

where ES is the pathway enrichment score, $T$ is the number of KOs that participate in the pathway and are found to be enriched in the assembled kelp bacterial genomes, $P$ is the hypergeometric $P$ value for each enriched KO, and $N$ is the total number of KOs in the specific KEGG pathway.

**Phylogenetic analysis of the assembled genomes.** We used all of the assembled genomes from the kelp microbiome to construct a phylogeny based on 36 phylogenetic marker proteins[67]. For each genome, proteins were aligned to individual COG proteins using BLASTP. The best hit to the protein that had at least 50% identity and covered more than 50% of the COG sequence was selected. For each COG, the selected proteins were aligned using MUSCLE (ref. 68) v3.8.31, and the individual alignments were concatenated into a single alignment. A phylogenetic tree was constructed with FastTree (ref. 69) v2.1.8 and visualized using ITOL (ref. 70).

**Data availability.** The metagenomic data generated in this study have been deposited in the BioProject database of Genbank under the accession number PRJNA357324. The MetaSort software and its source code have been deposited to Sourceforge (https://sourceforge.net/projects/metasort) and Github (https://github.com/jipeifeng/metasort). The authors declare that all other data supporting the findings of this study are available within the article and its Supplementary Information files.

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

## Acknowledgements

This work was supported by grants from the Strategic Priority Research Program of the Chinese Academy of Sciences [XDB13000000] and National Key R&D Program [2016YFC1200804] and the National Natural Science Foundation of China [91531306, 31301031].

## Author contributions

F.Z. conceived the project and designed the approach. P.J. and F.Z. designed and implemented the algorithm. Y.Z. and J.W. sequenced the samples and performed the experiments. P.J. performed the analysis. F.Z. and P.J. wrote the manuscript.
