## [Peer Review File · Nature Communications]

Reviewers' comments:

Reviewer #1 (Remarks to the Author):

The authors present an experimental and bioinformatics workflow for recovering uncultured genomes from microbial communities. The method reduces community complexity by using FCM to sequence "mini-metagenomes" of the original community and then combines these assemblies with high throughput sequencing in order to produce high quality genomes. I appreciate this methodology and can certainly see the power of the approach. Below I raise some concerns over the evaluations performed, especially in light of the additional expense and effort required to perform FCM and the subsequent sequencing of the mini-metagenomes.

Major comments:

- The presented simulation assumes perfect 'seed' contigs are available as input to the MGA algorithm. I believe this is essentially equivalent to assuming the BAF algorithm producing perfect 'target genome bins'. It does not appear that the quality of 'target genome bins' recovered from the BAF algorithm was evaluated. Can the authors please comment on this and discuss the impact and likelihood of BAF target genomes erroneously containing contigs from multiple genomes?
- For the simulation study, it is not clear to me what the clause "we set the recommended initial genome coverage to 30% in the downstream analyses" is aiming to convey. Can the authors provide any insights from the simulation on the required initial genome coverage for a near complete genome to be recovered? To what extent will this requirement be met by the cell sorting and BAF algorithm?
- The simulated community is relatively low complexity. In order to appreciate the MGA method, it would be helpful to know how available binning methods perform on this simulated dataset. Are similar quality genomes obtained using a binning method such as MetaBAT or MaxBin (i.e. as was demonstrated for the salivary metagenome)?
- The MGA algorithm was evaluated on a human salivary metagenome. However, this analysis is restricted to only enriched species with a coverage >30% in meta-S. This coverage information is only known by mapping to reference genomes, which will typically not be available. Why aren't the lower coverage meta-S genomes such as the 16% *Neisseria flavescens* genome considered? It would be informative to know how the MGA algorithm performs on these lower coverage genomes. Like the simulation study, these results appear to assume the "target genome bins" produced by the BAF algorithm are without error. Can the authors comment on why the MGA algorithm isn't evaluated in the context of the actual results produced by the BAF algorithm?
- While MetaBAT failed to recover most of the 7 salivary genomes obtained with the MGA algorithm, would it be a more equal to run MetaBAT on the contigs from both the meta-O and meta-S assemblies? Apply MetaBAT to just the meta-O assembly gives MGA a decisive advantage which makes it difficult to evaluate if the improved genomes are a result of the MGA algorithm itself or just the additional (high quality!) sequence data.
- MetaSort relies on FCM sorting and additional sequences to obtain the mini-metagenomes. Can the authors comment on the additional expense and time required to obtain the meta-S assemblies? I wonder if this additional expense was instead used to perform deeper metagenomic sequencing or to sequence multiple related metagenomic samples if metaSort would still outperform current methods such as MetaBAT.

- To better assess the performance of metaSort, it would be helpful to give the results of applying MetaBAT (or a similar binning method) to the kelp metagenome. Again, to evaluate the benefits of the MGA algorithm itself, both the meta-S and meta-O contigs should be provided to MetaBAT for binning. Perhaps the major benefit on this community is the FCM sorting and generation of the meta-S assemblies.

Minor comments:

- The reference "Chrisstian et al sequenced..." in the Introduction should be "Rinke et al...".
- I believe the abbreviation TGCG is only used once and never defined.
- It is unclear for the presented results if MetaBAT recovered additional genomes beyond the 7 shown in Supp. Table 2.
- It would help to briefly define NGA75 in the Results section since I don't believe this statistic is well known.

Reviewer #2 (Remarks to the Author):

This paper addresses the difficult problem of assembling metagenomic datasets. The described metaSort approach is two-pronged. Shotgun metagenome sequencing and assembly is used to create set of contigs called meta-O. A graph is constructed that is based on contig overlaps. These contigs are very fragmented due to complexity of metagenomes. Using size-selection, a mini-metagenome is isolated, applied using MDA, sequenced and assembled. The set of resulting contigs is called meta-S. Both sets of contigs are then clustered using different approaches. An algorithm called BAF is then used to compare the meta-S contigs with the meta-O contigs, and to bin contigs in meta-S in an accurate way. A second algorithm called MGA is then used to improve the binning and then to assemble individual genomes. This is a someone sophisticated that uses a number of major steps that operate on the contig connection graphs and aims at producing scaffolds. The algorithm attempts to compare strain-level variation using a logistic regression model.

The approach is applied to human oral and gut microbiomes, and the authors argue that the metaSort approach is able to capture strain-level diversity that is not visible using marker-gene techniques. The method is also applied to a kelp metagenomic sample. Here, three mini-metagenomes where sequenced and taxonomic analysis demonstrates a substantial reduction in the level of complexity hin these. 75 genomes were assembled at an average completeness of 78%, five genomes estimated to have 100% completely. This analysis was confirmed using PacBio reads: 90% of all PacBio reads mapped to a single scaffold, whereas only 4% mapped to different genomes.

This paper is a significant contribution to the problem of metagenome assembly. The meta-O + meta-S approach is very powerful and will surely be used by other researchers that need to assemble the most important genomes in their samples. The details of this paper have been worked out and reported in great detail, and is very well written (although some minor editing is still required).

Reviewer #3 (Remarks to the Author):

This is an intriguing and complex manuscript that describes a methodology, MetaSort, and associated

software for extracting genomes from metagenomes. The basic principle is to use flow cytometry to select subsets of cells based on some criteria e.g. size and then sequence these subsets separately following multiple displacement amplification (MDA). These subsets are then assembled and the resulting contigs compared to a metagenome of the original community. Various algorithms are then used to reconcile the subset contigs with the community metagenome and extract genome bins.

I thought that the combination of microfluidics and traditional metagenomics used here was intriguing but there were a number of methodological issues and technical inconsistencies in this manuscript that left me unsure as to how much of a genuine step forward this really represents. A lot of the complexity in their methods derives from artefacts associated with MDA. This results in only partial genomes with highly variable coverages being obtained from the subsets. However, since they are selecting a relatively large number of cells (tens of thousands) they could have just used low input DNA library preparation methods (as described in Rinke et al. PeerJ 2016: <https://peerj.com/articles/2486/>). These low input libraries whilst still having some biases give far better results in terms of genome coverage than MDA. Therefore, this would have simplified the entire computational strategy and obviated much of its complexity.

It may be because of these challenges associated with the MDA but I was not convinced that the results presented from the Kelp community were actually that good. They only obtained 75 genomes of varying levels of completion. The co-occurrence based binning methods dismissed in the introduction have achieved far better results than this, in some cases resolving thousands of genomes (see Brown et al. "Unusual biology across a group comprising more than 15% of domain Bacteria" Nature 2015). They have got the majority of the community but there was not attempt to assess, statistics could have been provided on fraction of reads in the metagenome mapping onto bins or number of core genes in the metagenome counted to resolve this.

Finally, the description of their algorithm is long and complex but contained some major technical inconsistencies. They describe their meta-O contig connection graph as a directed de Bruijn graph. A de Bruijn is formed of k-mers not contig sequences. I think what they constructed is a directed string graph. The description of the method was also rather low on references, the method for binning based on SVMs and sequence compositions is very similar to the strategies used in the PhyloPythia family of algorithms which should be cited.

REVIEWERS' COMMENTS:

Reviewer #1 (Remarks to the Author):

Thank you for the detailed and insightful responses to my previous comments. I plan to explore how metaSort can benefit my own research and believe it will be well received by the research community.

Reviewer #3 (Remarks to the Author):

The authors have addressed most of the issues raised by myself and the other reviewers. I do still have two minor comments though. Firstly, I was somewhat disappointed that they did not add some discussion of how the drawbacks of MDA could be mediated using low-DNA library preparation techniques (unless I missed it). In addition, if comparisons are going to be made to binning based on sample coverages, they could try CONCOCT in addition to MetaBat. CONCOCT has a more sophisticated variance model that may accommodate the noise in MDA coverage better, it can also bin contigs down to as short as 1kbp. Whilst I think these additional changes would improve the manuscript, I do feel that in its current form it is already acceptable for publication. The authors should be congratulated on producing a very intriguing paper.

Reviewers' comments:

Reviewer #1 (Remarks to the Author):

The authors present an experimental and bioinformatics workflow for recovering uncultured genomes from microbial communities. The method reduces community complexity by using FCM to sequence “mini-metagenomes” of the original community and then combines these assemblies with high throughput sequencing in order to produce high quality genomes. I appreciate this methodology and can certainly see the power of the approach. Below I raise some concerns over the evaluations performed, especially in light of the additional expense and effort required to perform FCM and the subsequent sequencing of the mini-metagenomes.

Response: We greatly appreciate the valuable comments from the reviewer. In this revised manuscript, we thoroughly revised the manuscript and added more detailed analyses on both simulated and real data sets. Please refer to the following responses for details.

Major comments:

- The presented simulation assumes perfect ‘seed’ contigs are available as input to the MGA algorithm. I believe this is essentially equivalent to assuming the BAF algorithm producing perfect ‘target genome bins’. It does not appear that the quality of ‘target genome bins’ recovered from the BAF algorithm was evaluated. Can the authors please comment on this and discuss the impact and likelihood of BAF target genomes erroneously containing contigs from multiple genomes?

Response: We thank the reviewer for pointing this out. To ensure the reliability of ‘target genome bins’, we have used very strict parameters in the BAF algorithm, including (i) The minimum contig length used for binning should be longer than 5 Kbp; (ii) The abundance information of meta-S contigs was calculated based on meta-O read pairs and was then used to improve the binning accuracy; (iii) When merging multiple bins on the graph, these bins should have one and only have one path connecting each other.

The simulation study presented in our manuscript was designed to test the performance of the MGA algorithm based on partial genome sequences. Considering

that current available simulation tools cannot mimic metagenomic data sets generated by MDA reaction, we cannot evaluate the binning accuracy of the BAF algorithm using simulated meta-S data sets. However, we calculated the contamination rate after binning the ‘perfect’ seed contigs and their corresponding meta-O contigs, and found that the average contamination rate was 0.61% across 99% of the recovered genomes. Please refer to Suppl. Fig. 6 for details. In addition, we further evaluated the accuracy of the BAF algorithm using the two real metagenomic datasets (human salivary and gut). After performing BAF on the salivary data set, 71.4% of the recovered genomes presented a contamination rate of zero and all other genomes had a contamination rate below 1.75%, except one was 53.37%. We have checked the highly contaminated bin and found that the contamination was caused by introducing genomes presenting similar sequences composition and abundance. These highly contaminated bins could easily be distinguished and discarded by CheckM. Regarding to the gut data set, the recovered bins also exhibited a low level of contamination rate, of which all target genome bins was below 4%, except one was 21%.

- For the simulation study, it is not clear to me what the clause “we set the recommended initial genome coverage to 30% in the downstream analyses” is aiming to convey. Can the authors provide any insights from the simulation on the required initial genome coverage for a near complete genome to be recovered? To what extent will this requirement be meet by the cell sorting and BAF algorithm?

Response: The initial genome coverage varies depending on the complexity and composition of metagenomic samples. There are two reasons that we set the recommended initial genome coverage to 30% for downstream analyses. Firstly, the average genome coverage of a typical MDA reaction was estimated to be 40% (*Rinke C, et al. Nature 499, 431-437, 2013.*). Secondly, a larger initial genome coverage will generally get a better combined assembly. The meta-S contigs randomly located on the target genome, will act as highly confident landmarks on the meta-O contig connection graph and will increase the probability of recovering nearby target contigs by using MGA. To make a balance of sensitivity and specificity, we used this initial genome coverage (30%) as a default threshold in our metaSort approach. However, users can set different thresholds for the initial genome coverage when they run the metaSort pipeline.

As suggested, we tried even lower initial genome coverage (19%~30%) in this revision, and found that both the contig length and genome coverage were significantly improved for these genomes (Suppl. Table 3). Specifically, the average genome coverage increased from 24.6% to 79.0%. But this improvement was obtained at the sacrifice of purity with contamination rate ranging from 8.3% to 26.4%, indicating that low initial genome coverage may introduce contaminated sequences from other genomes.

- The simulated community is relatively low complexity. In order to appreciate the MGA method, it would be helpful to know how available binning methods perform on this simulated dataset. Are similar quality genomes obtained using a binning method such as MetaBAT or MaxBin (i.e. as was demonstrated for the salivary metagenome)?

Response: In this revised manuscript, we have compared metaSort with other binning methods. Specifically, we applied MetaBAT and MaxBin on the simulated data set and compared their performance with metaSort. We firstly compared the recovered genome coverage. As shown in Suppl. Fig. 5a, metaSort exhibited a substantial improved genome coverage over the other two binning methods ($P < 1 \times 10^{-10}$, *t*-test). The median genome coverage was 91%, 24% and 78% for metaSort, metaBAT and MaxBin, respectively. The main reason responsible for such difference was primarily due to short contigs, where metaBAT and MaxBin discarded short contigs (below 1.5 Kbp) during the binning process. In contrast, these short contigs could be recovered through traversing the contig connection graph as implemented in the MGA algorithm. We further compared their performance on the assembled contig length. As shown in Suppl. Fig. 5b, metaSort showed considerably improved N50 length compared with the other two binning methods. The average N50 length were 37- and 45-fold longer than that of MaxBin and metaBAT, respectively.

- The MGA algorithm was evaluated on a human salivary metagenome. However, this analyses is restricted to only enriched species with a coverage >30% in meta-S. This coverage information is only known by mapping to reference genomes, which will typically not be available. Why aren't the lower coverage meta-S genomes such as the 16% *Neisseria flavescens* genome considered? It would be informative to know how the MGA algorithm performs on these lower coverage genomes. Like the simulation study, these results appear to assume the "target genome bins" produced by the BAF algorithm are without error. Can the authors comment on why the MGA algorithm isn't evaluated in the context of the actual results produced by the BAF algorithm?

Response: We thank the reviewer for pointing this out. In the previous version, we did not consider *Neisseria flavescens* for assembly because the genome coverage of this species was of low-abundance both in meta-S (16%) and meta-O (22%). As suggested, in this revised manuscript, we tested the MGA algorithm on lower coverage meta-S genomes (19%~30%). Results showed that although both the contig length and genome coverage exhibited considerable enhancement, the contamination rate also increased, indicating that very low initial genome coverage (<20%) could not provide sufficient information but introduce assembly errors.

In this revised manuscript, we added the accuracy evaluation of the BAF algorithm on both the human salivary and gut metagenomic datasets. After performing BAF on the salivary dataset, 71.4% of the recovered genomes presented a contamination rate of zero and all other genomes had a contamination rate below 1.75%, except one was 53.37%. As expected, the target genome bins obtained from the gut metagenome

dataset also exhibited a low level of contamination rate. The contamination rate for all target genome bins was below 4%, except one was 21%.

- While MetaBAT failed to recover most of the 7 salivary genomes obtained with the MGA algorithm, would it be a more equal to run MetaBAT on the contigs from both the meta-O and meta-S assemblies? Apply MetaBAT to just the meta-O assembly gives MGA a decisive advantage which makes it difficult to evaluate if the improved genomes are a result of the MGA algorithm itself or just the additional (high quality!) sequence data.

Response: We thank the reviewer's suggestion. In this revised manuscript, we applied metaBAT on the assemblies from both meta-O and meta-S. The quality of the 7 salivary genomes after performing metaSort and metaBAT were evaluated and compared. As shown in Supplementary Table 2, the binning results of metaBAT were indeed improved after combining the assemblies of meta-O and meta-S. However, metaSort exhibited much better performance on genome coverage (an average of 75.0%), sequence length (an average NGA50 length of 27 Kbp) and contamination rate (an average of 7.2%) compared with those of metaBAT (with genome coverage at an average of 58.1%, NGA50 length of 21 Kbp and contamination rate of 26.0%). The high contamination rate of metaBAT was mainly due to lacking of sequencing abundance information where the unevenness of MDA amplification masked the true abundance of the genomes in meta-S. All these results indicate that the improved genome quality should be attributed to the MGA algorithm rather than the incorporation of the meta-S data.

- MetaSort relies on FCM sorting and additional sequences to obtain the mini-metagenomes. Can the authors comment on the additional expense and time required to obtain the meta-S assemblies? I wonder if this additional expense was instead used to perform deeper metagenomic sequencing or to sequences multiple related metagenomic samples if metaSort would still outperform current methods such as MetaBAT.

Response: FCM sorting and MDA reaction can be done within one or two days. For example, we spent 5 hours on kelp sample preparation and FCM sorting (4 subsets), and spent 9 hours on cell lysis and MDA reaction. Regarding to expense, it is usually no more than \$80 per sample with four subsets (\$40 for FCM sorting and \$10 for one MDA reaction, \$40+\$10×4).

Considering the complexity of metagenomic samples, high-abundance bacteria have higher probability to be sequenced compared with moderate- or low-abundance species, and thus capturing low-abundance species is still difficult even using deeper metagenomic sequencing. Furthermore, deep metagenomic sequencing does not necessarily improve the metagenome assembly, which was demonstrated in various metagenomic studies. For instance, in the discussion part, we mentioned that "Howe *et al* performed a digital normalization on the HGMC dataset and found that 60% of

the total reads represented only 2% of captured bacteria”, which had no positive effect on final assembly. But instead, these “useless” data complicated the assembly graph and led to highly fragmented contigs that were difficult to be recovered by binning methods.

- To better assess the performance of metaSort, it would be helpful to give the results of applying MetaBAT (or a similar binning method) to the kelp metagenome. Again, to evaluate the benefits of the MGA algorithm itself, both the meta-S and meta-O contigs should be provided to MetaBAT for binning. Perhaps the major benefit on this community is the FCM sorting and generation of the meta-S assemblies.

Response: We thank the reviewer for pointing this out. In this revised manuscript, we applied metaBAT on the kelp metagenomic assemblies from both meta-O and meta-S. Then, the bins were evaluated using CheckM (Suppl. Fig. 13). As a result, we recovered 32 genomes by metaBAT, with contamination rate varying from 0% to 599% (average of 35%). In contrast, by applying metaSort, we successfully reconstructed 75 genomes, with the contamination rate varying from 0 to 10% (average of 3.4%). Furthermore, recovered genomes of metaSort exhibited considerably improved contig N50 compared with those of metaBAT.

Minor comments:

- The reference “Chrisstian et al sequenced...” in the Introduction should be “Rinke et al...”.

Response: The reference was updated.

- I believe the abbreviation TGCG is only used once and never defined.

Response: We thank the reviewer for pointing it out. We have changed “TGCG” to “target genome contig graph”.

- It is unclear for the presented results if MetaBAT recovered additional genomes beyond the 7 shown in Supp. Table 2.

Response: We have added the results of additional genomes in the Supplementary Table 3.

- It would help to briefly define NGA75 in the Results section since I don’t believe this statistic is well known.

Response: We have added the definition of NGA75 in the Results.

Reviewer #2 (Remarks to the Author):

This paper addresses the difficult problem of assembling metagenomic datasets. The described metaSort approach is two-pronged. Shotgun metagenome sequencing and assembly is used to create set of contigs called meta-O. A graph is constructed that is

based on contig overlaps. These contigs are very fragmented due to complexity of metagenomes. Using size-selection, a mini-metagenome is isolated, applied using MDA, sequenced and assembled. The set of resulting contigs is called meta-S. Both sets of contigs are then clustered using different approaches. An algorithm called BAF is then used to compare the meta-S contigs with the meta-O contigs, and to bin contigs in meta-S in an accurate way. A second algorithm called MGA is then used to improve the binning and then to assemble individual genomes. This is a someone sophisticated that uses a number of major steps that operate on the contig connection graphs and aims at producing scaffolds.

The algorithm attempts to compare strain-level variation using a logistic regression model.

The approach is applied to human oral and gut microbiomes, and the authors argue that the metaSort approach is able to capture strain-level diversity that is not visible using marker-gene techniques.

The method is also applied to a kelp metagenomic sample. Here, three mini-metagenomes were sequenced and taxonomic analysis demonstrates a substantial reduction in the level of complexity in these. 75 genomes were assembled at an average completeness of 78%, five genomes estimated to have 100% completely. This analysis was confirmed using PacBio reads: 90% of all PacBio reads mapped to a single scaffold, whereas only 4% mapped to different genomes.

This paper is a significant contribution to the problem of metagenome assembly. The meta-O + meta-S approach is very powerful and will surely be used by other researchers that need to assemble the most important genomes in their samples. The details of this paper have been worked out and reported in great detail, and is very well written (although some minor editing is still required).

Response: We greatly appreciate the valuable comments from the reviewer. In this revised manuscript, we thoroughly revised the manuscript.

Reviewer #3 (Remarks to the Author):

This is an intriguing and complex manuscript that describes a methodology, MetaSort, and associated software for extracting genomes from metagenomes. The basic principle is to use flow cytometry to select subsets of cells based on some criteria e.g. size and then sequence these subsets separately following multiple displacement amplification (MDA). These subsets are then assembled and the resulting contigs compared to a metagenome of the original community. Various algorithms are then used to reconcile the subset contigs with the community metagenome and extract genome bins.

I thought that the combination of microfluidics and traditional metagenomics used here was intriguing but there were a number of methodological issues and technical inconsistencies in this manuscript that left me unsure as to how much of a genuine step forward this really represents. A lot of the complexity in their methods derives from artefacts associated with MDA. This results in only partial genomes with highly variable coverages being obtained from the subsets. However, since they are selecting a relatively large number of cells (tens of thousands) they could have just used low input DNA library preparation methods (as described in Rinke et al. PeerJ 2016: <https://peerj.com/articles/2486/>). These low input libraries whilst still having some biases give far better results in terms of genome coverage than MDA. Therefore, this would have simplified the entire computational strategy and obviated much of its complexity.

Response: We greatly appreciate these valuable and insightful comments from the reviewer. We agree with the reviewer that newly developed library preparation methods with ultra-low DNA input will avoid MDA biases and thus will undoubtedly enhance the application of our metaSort approach.

It may be because of these challenges associated with the MDA but I was not convinced that the results presented from the Kelp community were actually that good. They only obtained 75 genomes of varying levels of completion. The co-occurrence based binning methods dismissed in the introduction have achieved far better results than this, in some cases resolving thousands of genomes (see Brown et al. “Unusual biology across a group comprising more than 15% of domain Bacteria” Nature 2015). They have got the majority of the community but there was not attempt to assess, statistics could have been provided on fraction of reads in the metagenome mapping onto bins or number of core genes in the metagenome counted to resolve this.

Response: We greatly appreciate the valuable comments from the reviewer. As suggested, we mapped the sequencing reads to the 75 recovered genomes, and found that 93% of total reads could be mapped to these recovered genomes. This strongly indicates that metaSort has captured the majority of the kelp bacterial community by using the combination of microfluidics and traditional metagenomics.

We agree with the reviewer that the co-occurrence-based binning methods work remarkably well when multiple samples are available. However, these approaches still face some inherent challenges: (i) sequencing and computing cost will be high with the increasing number of samples; (ii) the efficiency of co-occurrence-based methods is dependent on the complexity and heterogeneity of environmental samples. In addition, it is not always possible to obtain such a large number of samples (usually between 20 to 50); (iii) genomes that have intra-species variations but low abundance are generally assembled into small fragments, which are not applicable to most binning methods.

MetaSort, however, outperforms these methods for the following reasons: (i) it employs meta-S to reduce the complexity of metagenomic samples and to recover low-abundance species; (ii) it does not require a large number of samples for constructing co-occurrence profiles and thus will significantly reduce both computing and sequencing cost; (iii) it takes advantage of the meta-O contig connection graph to recover short target contigs, which are overlooked by most current binning approaches. Moreover, by traversing the target genome contig connection graph, the order of target contigs can be easily determined, and the sequences of these ordered contigs can be merged into long scaffolds.

Finally, the description of their algorithm is long and complex but contained some major technical inconsistencies. They describe their meta-O contig connection graph as a directed de Bruijn graph. A de Bruijn is formed of k-mers not contig sequences. I think what they constructed is a directed string graph. The description of the method was also rather low on references, the method for binning based on SVMs and sequence compositions is very similar to the strategies used in the PhyloPythia family of algorithms which should be cited.

Response: We thank the reviewer for pointing it out. We have revised the method part and changed “directed de Bruijn graph” to “directed string graph”. The mentioned papers have been cited.

Reviewers' comments:

Reviewer #3 (Remarks to the Author):

The authors have addressed most of the issues raised by myself and the other reviewers. I do still have two minor comments though. Firstly, I was somewhat disappointed that they did not add some discussion of how the drawbacks of MDA could be mediated using low-DNA library preparation techniques (unless I missed it). In addition, if comparisons are going to be made to binning based on sample coverages, they could try CONCOCT in addition to MetaBat. CONCOCT has a more sophisticated variance model that may accommodate the noise in MDA coverage better, it can also bin contigs down to as short as 1kbp. Whilst I think these additional changes would improve the manuscript, I do feel that in its current form it is already acceptable for publication. The authors should be congratulated on producing a very intriguing paper.

Response: We greatly appreciate these valuable and insightful comments from the reviewer. We have added the discussion on using low-DNA library preparation techniques to overcome the drawbacks of MDA in the introduction (please see line 77 for detail).

As suggested, we have performed additional evaluations on metaSort and CONCOCT by using the simulated, salivary and kelp metagenomics datasets. As a result, metaSort outperformed CONCOCT in terms of contig length, genome coverage and contamination rate across all of these datasets (please see Suppl. Fig. 5 and 14 for detail).